# Thinned Mean Field Langevin Dynamics

**Zonghao Chen** [1]   **Heishiro Kanagawa** [2]   **François-Xavier Briol** [1]   **Chris J. Oates** [2 3 *]   **Lester Mackey** [4 *]

## Abstract

Several important learning tasks can be formulated as minimizing an entropy-regularized objective over an appropriate space of probability distributions. Mean-field Langevin dynamics (MFLD) facilitate computation in this general context, casting the minimizer as the invariant distribution of a McKean–Vlasov process, which can be numerically discretized using $N$ particles and thus simulated. However, simulating this interacting particle system has computational complexity of order $N^2$. Motivated by recent research into *kernel thinning*, we propose KT-MFLD, in which each particle interacts only with a thinned particle coreset of size $\mathcal{O}(N^{\frac{1}{2}})$. KT-MFLD thus reduces the computational complexity to order $N^{\frac{3}{2}}$ while, under mild regularity conditions, achieving the same convergence guarantees (up to logarithmic factors) as MFLD. Our theoretical analysis is empirically confirmed on tasks including the training of student-teacher neural networks, quantization with maximum mean discrepancy, and computation of predictively-oriented posteriors in a post-Bayesian framework.

## 1. Introduction

Arguably the most popular regularizer in probabilistic machine learning is *entropy*; several diverse tasks can be formulated as finding the distribution $\pi$ which minimizes an entropy-regularized functional [79, 24, 16];

$$\mathscr{F}(\mu) := F(\mu) - \sigma \mathrm{Ent}(\mu). \tag{1}$$

Here, $\mathrm{Ent}(\mu) = -\int \log \mu(\boldsymbol{x})\mu(\mathrm{d}\boldsymbol{x})$ denotes the differential entropy, and $\sigma > 0$ controls the strength of entropic regularization. Here we overload notation so that the measure $\mu$ and its density are represented by the same symbol.

---
[*]Equal contribution  [1]University College London [2]Newcastle University [3]The Alan Turing Institute [4]Microsoft Research New England.  Correspondence to: Zonghao Chen <zonghao.chen.22@ucl.ac,uk>.

*Proceedings of the $43^{rd}$ International Conference on Machine Learning*, Seoul, South Korea. PMLR 306, 2026. Copyright 2026 by the author(s).

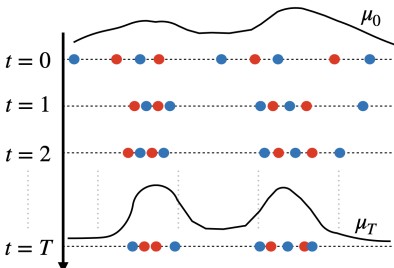

Figure 1. *Illustration of* KT-MFLD. In MFLD, all $N$ particles interact pairwise at each iteration $t \in \{0, 1, \ldots, T\}$, resulting in a cost of $\Omega(N^2)$. In contrast, in KT-MFLD, all $N$ particles (●; ●) interact only with a selected subset of $M$ particles (●), chosen via kernel thinning, leading to a reduced cost of $\mathcal{O}(N^{3/2})$.

In settings where $F$ is linear with respect to $\mu$, i.e. $F(\mu) = \int V(\boldsymbol{x})\mathrm{d}\mu(\boldsymbol{x})$, the minimizer $\pi$ can be written up to its normalization constant as $\pi(\boldsymbol{x}) \propto \exp(-V(\boldsymbol{x})/\sigma)$ [45, 63]. In contrast, when $F$ is *nonlinear*, no such explicit characterization of $\pi$ is available, precluding standard sampling methods such as Markov chain Monte Carlo [64]. Prominent examples of the above formulation with *nonlinear $F$* include the optimal mean-field distributions of two-layer infinitely wide neural networks [25, 42, 72], quantization with maximum mean discrepancy [3, 20, 22], emerging post-Bayesian frameworks for inference [85, 60, 74], trajectory inference [26, 40], and sparse inverse problems [10].

One popular approach in the nonlinear setting is mean-field Langevin dynamics (MFLD), a descent scheme that minimizes $\mathscr{F}$ in the *steepest* direction over $\mathcal{P}_2(\mathbb{R}^d)$, the space of probability distributions with finite second moment, with respect to the Wasserstein-2 metric [28, 42, 79, 24]. MFLD can be interpreted via its dual formulation as a McKean–Vlasov process; given an initial distribution $\mu_0 \in \mathcal{P}_2(\mathbb{R}^d)$,

$$\mathrm{d}X_s = -\nabla F'[\mu_s](X_s)\mathrm{d}s + \sqrt{2\sigma}\,\mathrm{d}B_s, \quad s > 0, \tag{2}$$

where $\mu_s = \mathrm{Law}(X_s)$, $F'[\mu_s] : \mathbb{R}^d \to \mathbb{R}$ is the *first variation* of $F$ at $\mu_s$ [73, Chapter 7], $\nabla$ denotes the Euclidean gradient, and $B_s$ is a standard Brownian motion on $\mathbb{R}^d$.

For linear $F$, the first variation $F'[\mu_s](X_s)$ is independent of $\mu_s$, and we recover Langevin dynamics, which can be simulated using a single particle [33]. In contrast, for nonlinear $F$ the first variation $F'[\mu_s](X_s)$ depends on $\mu_s$ as well as on $X_s$. Since $\mu_s$ is intractable, Eq. (2) cannot be directly simulated. A popular solution is to approximate Eq. (2)

using a system of $N$ interacting particles $\mathscr{X}_s := \{\boldsymbol{x}_s^{(i)}\}_{i=1}^N$ with associated empirical measure $\mu_{\mathscr{X}_s} = \frac{1}{N} \sum_{i=1}^N \delta_{\boldsymbol{x}_s^{(i)}}$, replacing the population-level drift $\nabla F'[\mu_s]$ by its empirical counterpart $\nabla F'[\mu_{\mathscr{X}_s}]$. This interacting particle system can be simulated, for example using the Euler–Maruyama method at discrete time steps $t = \{0, \dots, T\}$, and, under mild conditions on $F$, it has been proven that the empirical distribution $\mu_{\mathscr{X}_t}$ converges to the minimizer $\pi$ in Wasserstein-2 distance, as both $T, N \to \infty$ [24, 79, 65, 68].

Unfortunately, simulating the state $\boldsymbol{x}_{t+1}^{(i)}$ of the $i$-th particle at time $t+1$ requires computing interaction terms involving all of the $N$ particles at time $t$. This results in an $\Omega(N)$ cost per particle and an overall $\Omega(N^2)$ cost per iteration. This limitation becomes particularly severe when the target distribution $\pi$ is high-dimensional and multimodal, as representing such distributions typically requires a large number of particles. Such computational constraints often force practitioners to work with relatively small particle populations. For instance, in mean-field neural networks [67] and post-Bayesian inference [74], the number of particles is typically restricted to the order of $10^2$. Our experiments in Section 4 demonstrate that increasing the number of particles to the order of $10^3$ consistently leads to improved performance, highlighting the importance of more scalable particle-based simulations [44].

To reduce the $\Omega(N^2)$ per-iteration cost, we propose a new descent scheme that, at each iteration $t$, selects a representative subset $\{\bar{\boldsymbol{x}}_t^{(i)}\}_{i=1}^M \subset \{\boldsymbol{x}_t^{(i)}\}_{i=1}^N$ of size $M = \mathcal{O}(N^{1/2})$. We refer to this selection procedure as *thinning*. Each particle then interacts only with these $M$ representative particles, reducing the per-iteration computational complexity from $\Omega O(N^2)$ to $\mathcal{O}(N^{3/2})$. The effectiveness of this approach depends on how the representative particles are selected. For this work, we exploit a new near-linear time algorithm called KT-SPLIT-Compress [76] and we call the corresponding descent scheme KT-MFLD.

Our main theoretical contribution is to establish sufficient conditions under which KT-MFLD achieves accuracy comparable to MFLD, measured by the discrepancy between $\mu_{\mathscr{X}_T}$ and target distribution $\pi$. We further demonstrate these conditions are satisfied in the applications of MFLD considered in this work. In particular, for a fixed large $T$ and small step size, the finite-particle approximation error of KT-MFLD scales as $\mathcal{O}(N^{-1}(\log N)^2)$. This matches the approximation error of the original MFLD up to the logarithmic factor in $N$. By contrast, if the representative subset $\{\bar{\boldsymbol{x}}_t^{(i)}\}_{i=1}^M$ was selected via random subsampling at each iteration, referred to as Random-MFLD, the resulting finite-particle approximation error would deteriorate to $\Omega(N^{-\frac{1}{2}})$. Section 4 presents empirical results verifying that KT-MFLD outperforms Random-MFLD and other coreset-related acceleration methods [44] over a range of tasks for

which MFLD is commonly used.

**Notations:** Boldface symbols (e.g. $\boldsymbol{x}$) are used exclusively to denote particles, while all remaining variables are written in standard font. $\|\cdot\|$ without any subscripts denotes the Euclidean norm. For a bivariate differentiable function $q : \mathbb{R}^d \times \mathbb{R}^d \to \mathbb{R}$, we write $\nabla_1 q(\boldsymbol{x}, \boldsymbol{x}') \in \mathbb{R}^d$ (resp. $\nabla_2 q(\boldsymbol{x}, \boldsymbol{x}')$) as the gradient with respect to the first (resp. second) argument and $\mathbf{H}_1 q(\boldsymbol{x}, \boldsymbol{x}') \in \mathbb{R}^{d \times d}$ (resp $\mathbf{H}_2 q(\boldsymbol{x}, \boldsymbol{x}')$) as the Hessian with respect to the first (resp. second) argument. For a matrix $A \in \mathbb{R}^{d \times d}$, $\|A\|_{\mathrm{op}}$ denotes its spectral/operator norm. For a Hilbert space $\mathcal{H}$, we write $\mathcal{H}^{\otimes d}$ for its $d$-fold tensor product. For $\mu \in \mathcal{P}_2(\mathbb{R}^d)$, we write $\mu^{\otimes N} \in \mathcal{P}_2((\mathbb{R}^d)^N)$ for the $N$-fold product measure. We use the standard asymptotic notation $\mathcal{O}(\cdot), \Omega(\cdot)$, and $\Theta(\cdot)$ to denote upper, lower, and matching upper–lower bounds, respectively, up to multiplicative constants. $\tilde{\mathcal{O}}(f(n))$ means $\mathcal{O}(f(n))$ up to polylog factors in $n$.

## 2. Background

We first discuss background material and related work. For the purposes of this work, consider the broad class of nonlinear functionals of the form $F(\mu) = F_0(\mu) + \frac{\varsigma}{2}\mathbb{E}_\mu[\|\boldsymbol{x}\|^2]$ for $\varsigma > 0$, where

$$F_0(\mu) = \mathbb{E}_{u \sim \rho}\left[R_1\left(\int q_1(u, \boldsymbol{x})\mathrm{d}\mu(\boldsymbol{x})\right)\right]$$
$$+ \frac{1}{2}\iint q_2(\boldsymbol{x}, \boldsymbol{x}^\circ)\mathrm{d}\mu(\boldsymbol{x})\mathrm{d}\mu(\boldsymbol{x}^\circ), \quad (3)$$

for differentiable $R_1 : \mathbb{R} \to \mathbb{R}$, $q_1 : \mathbb{R}^{d_u} \times \mathbb{R}^d \to \mathbb{R}$, $q_2 : \mathbb{R}^d \times \mathbb{R}^d \to \mathbb{R}$, and $u \in \mathbb{R}^{d_u}$. In addition, we assume that $\rho$ is some distribution on the latent variable $u$, that $R_1$ is convex, and that $q_2$ is symmetric ($q_2(\boldsymbol{x}, \boldsymbol{x}^\circ) = q_2(\boldsymbol{x}^\circ, \boldsymbol{x})$) and positive definite [78, Def. 4.15]. This form of $F$ encompasses the vast majority of existing applications of MFLD, as reviewed below; more general forms of $F$ are discussed in Appendix A. The minimizer $\pi$ is unique and absolutely continuous with respect to the Lebesgue measure under mild smoothness and boundedness conditions on $R_1, q_1, q_2$ [42, Prop. 2.5].

**Mean-Field Langevin Dynamics** In the above setting, the velocity field governing the particle evolution of MFLD takes the form:

$$\nabla F'[\mu](\boldsymbol{x}) = \nabla F_0'[\mu](\boldsymbol{x}) + \varsigma\boldsymbol{x}$$
$$\nabla F_0'[\mu](\boldsymbol{x}) = \mathbb{E}_{u \sim \rho}\left[R_1'\left(\int q_1(u, \boldsymbol{x}^\circ)\mathrm{d}\mu(\boldsymbol{x}^\circ)\right)\nabla_2 q_1(u, \boldsymbol{x})\right]$$
$$+ \int \nabla_1 q_2(\boldsymbol{x}, \boldsymbol{x}^\circ)\mathrm{d}\mu(\boldsymbol{x}^\circ). \quad (4)$$

As a result, the McKean-Vlasov interpretation of MFLD in Eq. (2) simplifies to the following stochastic differential

equation: given an initial distribution $\mu_0 \in \mathcal{P}_2(\mathbb{R}^d)$,

$$\mathrm{d}X_s = -\{\nabla F_0'[\mu_s](X_s) + \zeta X_s\}\mathrm{d}s + \sqrt{2\sigma}\,\mathrm{d}B_s, \quad s > 0,$$

where $\mu_s = \mathrm{Law}(X_s)$. In practice, the above continuous-time dynamics is approximated by a system of interacting particles $\{x_t^{(i)}\}_{i=1}^N$, evolving over discrete time steps for a fixed time horizon $T \in \mathbb{N}$. Given an initial collection of particles $\mathscr{X}_0 = \{x_0^{(i)}\}_{i=1}^N$ sampled i.i.d from $\mu_0$, the update rule is given by

$$x_{t+1}^{(i)} \tag{5}$$
$$= x_t^{(i)} - \gamma\{\nabla F_0'[\mu_{\mathscr{X}_t}](x_t^{(i)}) + \zeta x_t^{(i)}\} + \sqrt{2\sigma\gamma}\,\xi_t^{(i)}$$

for each time $t \in \{0, \dots, T\}$ and particle $i \in \{1, \dots, N\}$. Here, $\gamma > 0$ is a fixed step size, and $\{\xi_t^{(i)}\}_{i=1}^N$ are $N$ i.i.d. samples from the unit Gaussian $\mathcal{N}(0, \mathrm{Id})$. We refer to the discrete-time particle system defined in Eq. (5) as MFLD, to distinguish it from MFLD defined in Eq. (2). For each particle $x_t^{(i)}$, the vector field $\nabla F_0'[\mu_{\mathscr{X}_t}](x_t^{(i)})$, as shown in Eq. (4), requires computing the integral with respect to an empirical distribution $\mu_{\mathscr{X}_t}$, which amounts to an empirical average over the set of $N$ particles $\{x_t^{(i)}\}_{i=1}^N$. Specifically,

$$\nabla F_0'[\mu_{\mathscr{X}_t}](x_t^{(i)})$$
$$= \mathbb{E}_{u \sim \rho}\left[ R_1'\left(\frac{1}{N}\sum_{j=1}^N q_1(u, x_t^{(j)})\right)\nabla_2 q_1(u, x_t^{(i)})\right]$$
$$+ \frac{1}{N}\sum_{j=1}^N \nabla_1 q_2(x_t^{(i)}, x_t^{(j)}). \tag{6}$$

Therefore, repeating this computation for all $N$ particles in the descent scheme of Eq. (5) results in a $\Omega(N^2)$ per-iteration computational cost.

*Additional Linear Terms:* The objective $F_0$ in Eq. (3) may additionally contain a linear term $\int q_3(x)\mathrm{d}\mu(x)$, which would contribute to an additional $\frac{1}{N}\sum_{j=1}^N \nabla q_3(x_t^{(j)})$ in the vector field in Eq. (6). Since this new term can be computed in $\mathcal{O}(N)$ time, which is dominated by the $\Omega(N^2)$ interaction term, we omit it from the presentation.

**Illustrative Applications** Here we highlight several representative applications of MFLD and their objectives $F_0$. Application 1 corresponds to the instantiation in which only the first term of $F_0$ is present, whereas Applications 2 and 3 correspond to the instantiations in which only the second term is present.

*1. Mean-Field Neural Networks:* Consider a mean-field two-layer neural network: $h_\mu(z) = \mathbb{E}_{x \sim \mu}[\Psi(z, x)]$ where $\Psi(z, x) = w_2 a(w_1^\top z)$ with input $z$ and parameters $x = (w_1, w_2)$. Here, $w_1$ and $w_2$ denote the first- and second-layer weights, and $a$ is the activation function. Given a

loss function $\ell : \mathbb{R} \times \mathbb{R} \to \mathbb{R}$ and a data distribution $\rho$ on $(z, y)$, the corresponding loss objective is $F_0(\mu) = \mathbb{E}_{(z,y) \sim \rho}[\ell(h_\mu(z), y)]$. Compared with Eq. (3), this setting involves only the first term with $q_1(u, x) = \Psi(z, x)$, and $R_1$ is the pointwise loss function: $R_1(t; u) := \ell(t, y)$ with $u = (z, y)$. The dynamics of Eq. (5) coincides with noisy gradient descent on a finite-width neural network of width $N$. This correspondence was established by [66, 72, 61, 25] and has recently been exploited in subsequent analyses [42, 17, 79, 24, 67, 68, 21] to derive non-asymptotic convergence guarantees for finite-width two-layer networks. In this paper, we consider an *online* setting in which, at each iteration, a fresh batch of samples is drawn from the teacher network to update each network particle; see Remark C.1 for further discussion.

*2. Quantization with Maximum Mean Discrepancy:* A recent line of work considers selecting a set of particles $\{x^{(i)}\}_{i=1}^N$ that best approximate a target distribution $\pi$ by minimizing the squared maximum mean discrepancy (MMD) with respect to a kernel $\varkappa$ [39]: $F_0(\mu) = \iint \varkappa(x, x^\circ)\mathrm{d}\mu(x)\mathrm{d}\mu(x^\circ) - 2\varkappa(x, y)\mathrm{d}\mu(x)\mathrm{d}\pi(y) + \mathrm{const}$. Disregarding the term that is linear in $\mu$, and comparing with Eq. (3), this setting involves only the second term with $q_2(x, x^\circ) = \varkappa(x, x^\circ)$. This framework has been explored in the context of generative modeling [3, 20], sampling from unnormalized distributions [50, 81], and numerical integration [86, 22].

*3. Predictively-Oriented (PrO) Posteriors:* An emerging direction in post-Bayesian methodology aims to learn a distribution $\mu$ over model parameters $x$ by matching the induced predictive distribution $\int P_x \mathrm{d}\mu(x)$ to a dataset $\{y_i\}_{i=1}^n \subset \mathcal{Y}$, where, for each $x$, $P_x$ denotes a conditional probability distribution on the observation space $\mathcal{Y}$ (e.g., a likelihood or generative model) indexed by some parameters $x$ [59, 43, 75, 62, 51, 74, 57]. This can be achieved using a proper scoring rule $S : \mathcal{P}_2(\mathbb{R}^d) \times \mathcal{Y} \to \mathbb{R}$, for example the log scoring rule $S(\nu, y) = -\log \nu(y)$ or the kernel scoring rule $S(\nu, y) = 2\int \varkappa(y, y')\mathrm{d}\nu(y') - \iint \varkappa(y', y'')\mathrm{d}\nu(y')\mathrm{d}\nu(y'')$ [38]. The *PrO posterior* is obtained as the solution to an optimization problem with $F_0(\mu) = \sum_{i=1}^n S(\int P_x \mathrm{d}\mu(x), y_i)$. For the kernel scoring rule, and disregarding the term that is linear in $\mu$, this setting involves only the second term with $q_2(x, x^\circ) = \iint \varkappa(y, y')\mathrm{d}P_x(y)\mathrm{d}P_{x^\circ}(y')$. PrO posteriors are attracting interest because they are capable of adapting to model misspecification while the standard Bayesian posterior is not [60]. However, their widespread adoption is currently limited by the lack of effective numerical methods compared to the standard Bayesian setting.

**Other Interacting Particle Systems** The update scheme of MFLD in Eq. (5) is the canonical interacting particle system for approximating $\pi$ [28], but other interacting particle sys-

tems have been proposed. For example, *Stein variational gradient descent* [82] and *kernel gradient descent* [16] are deterministic alternatives to MFLD. This paper focuses on MFLD, for which the sharpest results can be established, but our proposed thinning technique can be applied also to other interacting particle systems; we reserve a discussion for Appendix A. There also exist alternative acceleration strategies for interacting particle systems beyond subsampling the particles [54]; we provide a detailed discussion of these related approaches in Appendix A.

**Kernel Thinning** We have now concluded our presentation of MFLD and can discuss the main tool we will use to accelerate it. *Kernel thinning* aims to *thin* (i.e., subsample) a sequence of points $\{\boldsymbol{x}^{(i)}\}_{i=1}^{N}$ with size $N$ to a much smaller subset $\{\bar{\boldsymbol{x}}^{(i)}\}_{i=1}^{M}$ of size $M = \lceil\sqrt{N}\rceil$ points without degrading its quality [35, 34, 76, 53]. For a symmetric positive semi-definite kernel $k : \mathbb{R}^d \times \mathbb{R}^d \to \mathbb{R}$, its corresponding reproducing kernel Hilbert space (RKHS) $\mathcal{H}_k$ is a Hilbert space with inner product $\langle\cdot,\cdot\rangle_{\mathcal{H}_k}$ and norm $\|\cdot\|_{\mathcal{H}_k}$ [4], such that (i) $k(\boldsymbol{x},\cdot) \in \mathcal{H}_k$ for all $\boldsymbol{x} \in \mathbb{R}^d$, and (ii) the reproducing property holds, i.e. for all $f \in \mathcal{H}_k, \boldsymbol{x} \in \mathbb{R}^d, f(\boldsymbol{x}) = \langle f, k(\boldsymbol{x},\cdot)\rangle_{\mathcal{H}_k}$. The quality of the thinned coreset is evaluated by the integration error

$$\mathcal{E}(f) := \Big|\frac{1}{N}\sum_{i=1}^{N} f(\boldsymbol{x}^{(i)}) - \frac{1}{M}\sum_{i=1}^{M} f(\bar{\boldsymbol{x}}^{(i)})\Big|$$

for functions $f \in \mathcal{H}_k$. In this paper, among the family of kernel thinning algorithms, we are primarily interested in the KT-SPLIT-Compress($\delta$) algorithm proposed in [76], because of its near-linear $\mathcal{O}(N \log N)$ computational cost and strong theoretical guarantees:

**Proposition 2.1** (Adapted from Theorem 1 of [76]; see Appendix B.1). *Suppose that* $\kappa := \sup_{\boldsymbol{x}\in\mathbb{R}^d} k(\boldsymbol{x},\boldsymbol{x}) < \infty$. *Let* $\mathfrak{g} \in \mathbb{N}_0$. *Let* $\{\boldsymbol{x}^{(i)}\}_{i=1}^{N} \subset \mathbb{R}^d$ *be a point set and* $\{\bar{\boldsymbol{x}}^{(i)}\}_{i=1}^{M}$ *be the thinned output of* KT-SPLIT-Compress($\delta$) *with* $M = 2^{\mathfrak{g}}\lceil\sqrt{N}\rceil$ *and* $\delta \in (0,1)$. *Then, for any fixed* $f \in \mathcal{H}_k$,

$$\mathcal{E}(f) \leq \frac{2\|f\|_{\mathcal{H}_k}\sqrt{\kappa}}{M}\sqrt{\log_2(\tfrac{N}{M})\log\big(\tfrac{6M}{\delta}\log_2(\tfrac{N}{M})\big)\log(\tfrac{1}{\delta})},$$

*with probability at least* $1 - \delta$.

Here, $\mathfrak{g} \in \mathbb{N}_0$ is an *oversampling* parameter that determines the coreset size $M = 2^{\mathfrak{g}}\lceil\sqrt{N}\rceil$: larger values of $\mathfrak{g}$ lead to smaller integration error but results in higher computational cost in downstream tasks. In a special case of $\mathfrak{g} = 0$ such that $M = \lceil\sqrt{N}\rceil$, Proposition 2.1 implies that, for each $f \in \mathcal{H}_k$, the coreset achieves $\tilde{\mathcal{O}}(N^{-\frac{1}{2}})$ integration error. The additional logarithmic factor can be avoided by an appropriate choice of the oversampling parameter $\mathfrak{g} = \lceil\log_2\log N\rceil$, which yields a coreset of size $M = 2^{\mathfrak{g}}\lceil\sqrt{N}\rceil$ and achieves $\mathcal{O}(N^{-\frac{1}{2}})$ integration error.

Both are a substantial improvement over the $\Omega(N^{-\frac{1}{4}})$ integration error if the points $\{\bar{\boldsymbol{x}}^{(i)}\}_{i=1}^{M}$ are instead uniformly sampled from the original point set $\{\boldsymbol{x}^{(i)}\}_{i=1}^{N}$. The proof of Proposition 2.1 can be found in Appendix B.1.

Alternative thinning methods can also be considered, such as KT-Compress($\delta$) [76] which constructs a coreset of size $M = 2^{\mathfrak{g}}\lceil\sqrt{N}\rceil$ in near-linear time and provably controls integration error under conditions on $\max_i \|\boldsymbol{x}^{(i)}\|$. At present it is unclear whether a uniform-in-time bound on $\max_i \|\boldsymbol{x}_t^{(i)}\|$ can be derived for MFLD, but nevertheless we also include KT-Compress in our empirical assessment in Section 4. Other thinning (also known as *compression*) algorithms that provably achieve smaller integration error than uniform subsampling, include kernel thinning and KT-SPLIT (without Compress) [35, 34], Gram-Schmidt thinning [14], and stationary point methods [22]. Other works establish superior rates for idealized point sets but do not analyze algorithms that are guaranteed to construct such point sets [9, 58, 46, 86]. Some methods, like kernel herding [18] and sequential greedy algorithms [69, 80], demonstrate strong empirical performances without theoretical guarantees of outperforming uniform subsampling. In this paper, we focus on KT-SPLIT-Compress and KT-Compress due to their favorable *near-linear* computational cost. In contrast, all of the other aforementioned methods incur at least quadratic $\Omega(N^2)$ computational cost.

## 3. Thinned Mean Field Langevin Dynamics

Despite the broad interest in MFLD, its $\Omega(N^2)$ computational cost limits its widespread use on large-scale problems. We now propose a novel algorithm, called *kernel-thinned mean field Langevin dynamics* (KT-MFLD), which reduces this cost to $\mathcal{O}(N^{\frac{3}{2}})$ whilst maintaining the same convergence guarantees as the original MFLD up to logarithmic factors in $N$; see Theorem 3.3.

KT-MFLD closely resembles MFLD but, at each iteration, we first thin the set of $N$ particles into a smaller coreset of size $M = \lceil\sqrt{N}\rceil$ using KT-SPLIT-Compress($\delta$) with $\delta = \frac{(\log_2 N)^3}{N}$ and oversampling parameter $\mathfrak{g} = 0$, then replace the empirical average in the vector field of Eq. (5), originally taken over all $N$ particles, with an average over the thinned subset of particles. More precisely, given an initial collection of particles $\mathscr{X}_0$ and a time horizon $T \in \mathbb{N}$, for each time $t \in \{0,\ldots,T\}$ and for each particle $i \in \{1,\ldots,N\}$, the update rule is given by:

$$\boldsymbol{x}_{t+1}^{(i)} \tag{7}$$
$$= \boldsymbol{x}_t^{(i)} - \gamma\big[\nabla F_0'(\mu_{\bar{\mathscr{X}}_t})(\boldsymbol{x}_t^{(i)}) + \zeta\boldsymbol{x}_t^{(i)}\big] + \sqrt{2\sigma\gamma}\,\xi_t^{(i)}$$

where $\{\xi_t^{(i)}\}_{i=1}^{N} \sim \mathcal{N}(0,\mathrm{Id})$ and $\bar{\mathscr{X}}_t = \{\bar{\boldsymbol{x}}_t^{(i)}\}_{i=1}^{M}$ is obtained from thinning $\mathscr{X}_t$ using KT-SPLIT-Compress as just described. Compared with the original MFLD in Eq. (5),

the update scheme of KT-MFLD is governed by the new vector field $\nabla F_0'(\mu_{\bar{\mathscr{X}}_t})$ evaluated at the thinned empirical distribution $\mu_{\bar{\mathscr{X}}_t}$, rather than the empirical distribution made of the full set of particles. Specifically,

$$
\begin{aligned}
&\nabla F_0'(\mu_{\bar{\mathscr{X}}_t})(\boldsymbol{x}_t^{(i)}) \\
&= \mathbb{E}_{u \sim \rho}\left[ R_1'\left(\frac{1}{M}\sum_{j=1}^M q_1(u, \bar{\boldsymbol{x}}_t^{(j)})\right) \nabla_2 q_1(u, \boldsymbol{x}_t^{(i)})\right] \\
&\quad + \frac{1}{M}\sum_{j=1}^M \nabla_1 q_2(\boldsymbol{x}_t^{(i)}, \bar{\boldsymbol{x}}_t^{(j)}),
\end{aligned} \tag{8}
$$

now involves an empirical average over the thinned subset of size $M$, thereby reducing the per-iteration computational cost to $\mathcal{O}(N^{\frac{3}{2}})$. Since KT-SPLIT-Compress incurs a cost of $\tilde{\mathcal{O}}(N)$ and is executed only once per iteration, the total computational cost per iteration is now reduced to $\mathcal{O}(N^{\frac{3}{2}}) + \tilde{\mathcal{O}}(N) = \mathcal{O}(N^{\frac{3}{2}})$.

Next, we rigorously show that KT-MFLD in Eq. (7) retains nearly the same convergence guarantee as the original MFLD in Eq. (5) under the following regularity assumptions:

**Assumption 3.1** (Boundedness, Lipschitzness and Convexity). *There exists positive constants $Q_1, Q_2$, such that $\sup_{u,\boldsymbol{x}}|q_1(u,\boldsymbol{x})| \leq Q_1$, $\sup_{u,\boldsymbol{x}}\|\nabla_2 q_1(u,\boldsymbol{x})\| \leq Q_1$, $\sup_{u,\boldsymbol{x}}\|\mathbf{H}_2 q_1(u,\boldsymbol{x})\|_{\mathrm{op}} \leq Q_1$; and such that $\sup_{\boldsymbol{x},\boldsymbol{x}^\circ}|q_2(\boldsymbol{x},\boldsymbol{x}^\circ)| \leq Q_2$, $\sup_{\boldsymbol{x},\boldsymbol{x}^\circ}\|\nabla_1 q_2(\boldsymbol{x},\boldsymbol{x}^\circ)\| \leq Q_2$, $\sup_{\boldsymbol{x},\boldsymbol{x}^\circ}\|\mathbf{H}_1 q_2(\boldsymbol{x},\boldsymbol{x}^\circ)\|_{\mathrm{op}} \leq Q_2$. $R_1 : \mathbb{R} \to \mathbb{R}$ is convex and its derivative is $L_R$-Lipschitz, i.e., $|R_1'(y) - R_1'(y^\circ)| \leq L_R|y - y^\circ|$ for any $y, y^\circ \in \mathbb{R}$.*

**Assumption 3.2.** *The reproducing kernel $k : \mathbb{R}^d \times \mathbb{R}^d \to \mathbb{R}$ is bounded; i.e. $\kappa := \sup_{\boldsymbol{x} \in \mathbb{R}^d} k(\boldsymbol{x}, \boldsymbol{x}) < \infty$. Further, $q_1(u, \cdot) \in \mathcal{H}_k$ for each $u$ and $\nabla_1 q_2(\boldsymbol{x}, \cdot) \in \mathcal{H}_k^{\otimes d}$ for each $\boldsymbol{x}$, and $\mathcal{Q}_1 := \sup_u \|q_1(u, \cdot)\|_{\mathcal{H}_k} < \infty$, $\mathcal{Q}_2 := \sup_{\boldsymbol{x}} \|\nabla_1 q_2(\boldsymbol{x}, \cdot)\|_{\mathcal{H}_k^{\otimes d}} < \infty$.*

Assumption 3.1 provides sufficient conditions that imply the following key properties of $F_0$ (as established in Appendix B): (i) the vector field $\nabla F_0'$ is uniformly bounded and Lipschitz continuous (as established in Lemma B.3); (ii) the functional $F_0 : \mathcal{P}_2(\mathbb{R}^d) \to \mathbb{R}$ is convex (as established in Lemma B.4); and (iii) $F_0$ satisfies a leave-one-out stability property (as established in Lemma B.6). These three key properties align with the sufficient conditions used in state-of-the-art analyses of MFLD [68, Assumptions 1, 2, and 4]. Assumption 3.2, however, is a new assumption imposed in this paper such that $q_1$ and $\nabla q_2$ belong to appropriate RKHSs which ensure thinning properties of KT-SPLIT-Compress in Proposition 2.1 hold.

**Theorem 3.3** (Convergence of KT-MFLD). *Suppose Assumptions 3.1 and 3.2 hold. Let $\mu_t^{(N)} \in \mathcal{P}_2((\mathbb{R}^d)^N)$ be the joint distribution of $N$ particles from Eq. (7) at iteration*

$t$.[1] *Suppose the step size $\gamma \leq \frac{1}{8} \wedge \frac{1}{2\zeta}$. Suppose the initial $N$ particles are sampled i.i.d from $\mu_0 \in \mathcal{P}_2(\mathbb{R}^d)$ with $\mathrm{Ent}(\mu_0) < \infty$. Then there exist constants $C_{\mu_0}, C_\gamma, B, c_0$ and $C_3$ such that, for each $T \in \mathbb{N}$,*

$$
\begin{aligned}
&\frac{\sigma}{N}\mathrm{KL}\left(\mu_T^{(N)} \| \pi^{\otimes N}\right) \\
&\leq \exp(-T\bar{\alpha}\sigma\gamma)C_{\mu_0} + \frac{B}{N} + \frac{C_\gamma}{2\bar{\alpha}} + \frac{c_0(\log N)^3}{N\bar{\alpha}}, \quad (9)
\end{aligned}
$$

*where $\bar{\alpha} = \frac{\zeta}{2\sigma}\exp(-\frac{4C_3}{\zeta\sigma}\sqrt{\frac{2d}{\pi}})$. The constant $C_{\mu_0}$ depends only on the initial distribution $\mu_0$ and $C_\gamma = \mathcal{O}(\gamma\sigma d + \gamma^2)$ vanishes as the time step $\gamma$ decreases. The constants $B$, $c_0$, and $C_3$ depend only on the constants in Assumptions 3.1 and 3.2.*

The proof of Theorem 3.3 is in Appendix B.2, and is adapted from the state-of-the-art convergence result of standard MFLD by [68]. Theorem 3.3 shows that, for sufficiently large $N$ and $T$ and a sufficiently small step size $\gamma$, the particles $\{\boldsymbol{x}_T^{(i)}\}_{i=1}^N$ generated by KT-MFLD behave asymptotically as $N$ i.i.d. samples from the target $\pi$.

The main difference in our setting, from standard MFLD, is the presence of an additional error introduced by taking the empirical average over the *thinned coreset* when computing the vector field at each iteration. Compared with point 2 of Theorem 1 in [68],

$$
\frac{\sigma}{N}\mathrm{KL}\left(\mu_T^{(N)} \| \pi^{\otimes N}\right) \leq \exp(-T\bar{\alpha}\sigma\gamma)C_{\mu_0} + \frac{B}{N} + \frac{C_\gamma}{2\bar{\alpha}}.
$$

our convergence bound includes an extra $\mathcal{O}(N^{-1}(\log N)^3)$ term accounting for this thinning-induced approximation error. By contrast, if one were to randomly select $M = \lceil\sqrt{N}\rceil$ samples from the $N$ particles at each iteration, as in [79], the resulting upper bound would entail an extra term that scales as $\Omega(N^{-\frac{1}{2}})$ (cf. Remark B.8).

Therefore, for our KT-MFLD to achieve an error $\frac{1}{N}\mathrm{KL}(\mu_T^{(N)} \| \pi^{\otimes N}) \leq \varepsilon$, it suffices to choose the step size $\gamma = \mathcal{O}(\varepsilon)$, the number of iterations $T = \Omega(\log(\varepsilon^{-1})\varepsilon^{-1})$, and particle number $N = \tilde{\Omega}(\varepsilon^{-1})$. This yields a per-iteration computational complexity of $\tilde{\Theta}(\varepsilon^{-3/2})$. For the same choice of step size $\gamma$ and number of iterations $T$, standard MFLD requires $N = \Omega(\varepsilon^{-1})$ particles, while Random-MFLD requires $N = \Omega(\varepsilon^{-2})$ particles. Their resulting per-iteration complexities are therefore $\Omega(\varepsilon^{-2})$ and $\Omega(\varepsilon^{-3})$, respectively. We note, however, that the thinning-induced error term $c_0(\log N)^3/(N\bar{\alpha})$ suffers from a curse of dimensionality, since $\bar{\alpha}$ can be exponentially small in $d$ [23]. Consequently, the particle complexity of KT-MFLD hides an adverse dependence on the dimension. By contrast, in the original MFLD bound, $\bar{\alpha}$ enters only through the

---

[1]Note that $\mu_{\mathscr{X}_t}$ denotes the empirical distribution of $N$ particles at time $t$, while $\mu_t^{(N)}$ denotes the joint distribution on $(\mathbb{R}^d)^N$.

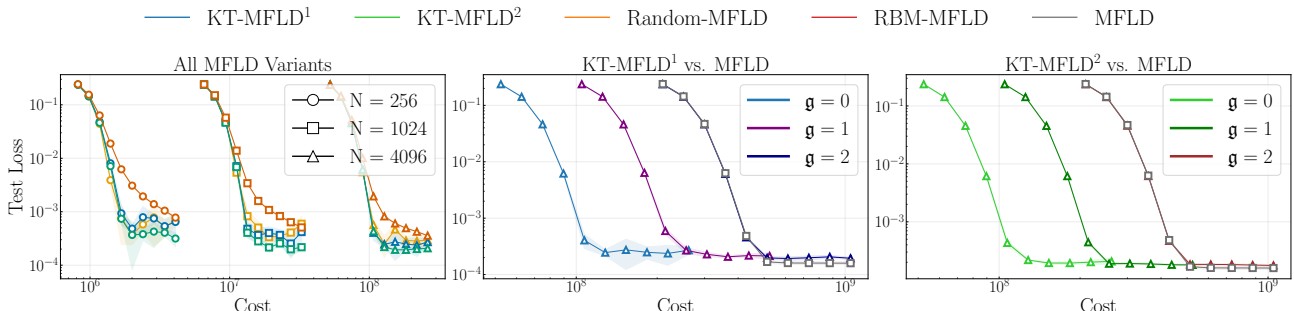

*Figure 2. Student-Teacher network.* **Left:** Comparison of KT-MFLD[1] and KT-MFLD[2] with all subsampling based acceleration baselines. **Middle:** Comparison of KT-MFLD[1] with original MFLD under varying $\mathfrak{g}$. **Right:** Comparison of KT-MFLD[2] with original MFLD under varying $\mathfrak{g}$. Results are averaged over 20 independent runs. Shaded areas represent $\pm 2\times$ standard error.

choice of step size and the resulting iteration complexity, rather than through the required number of particles [65, 68].

One closely related sub-sampling based acceleration technique is the random batch method (RBM-MFLD) [44], which is widely used for accelerating interacting particle systems in computational physics. In RBM-MFLD, the full particle system at each iteration $\{\boldsymbol{x}_t^{(i)}\}_{i=1}^N$ is randomly partitioned into batches of size $p$. The update scheme in Eq. (5) is then applied independently within each batch, so that the drift term in Eq. (6) only involves interactions among particles within the same batch. As a result, the per-iteration computational cost is reduced to $\mathcal{O}((N/p)\times p^2) = \mathcal{O}(Np)$. Although Corollary 3.1 of [44] provides a convergence guarantee, it relies on a strong contraction assumption that is not satisfied in our MFLD setting. Consequently, their associated theoretical batch-size prescriptions are not applicable here. In our experiments, we therefore choose the batch size of RBM-MFLD as $p = \lceil\sqrt{N}\rceil$ purely for the purpose of fair computational comparison. With this choice, the per-iteration cost of RBM-MFLD scales as $\mathcal{O}(N^{\frac{3}{2}})$ and matches that of Random-MFLD and KT-MFLD. RBM-MFLD coincides with the ensembling method proposed in Nitanda et al. [68] in the context of mean-field neural networks.

Next, we show that Assumptions 3.1 and 3.2 can indeed be satisfied under mild conditions in all the practical applications considered in Section 2.

**Remark 3.4** (Sufficient conditions for Assumptions 3.1 and 3.2).
*1. Mean-Field Two-Layer Neural Networks: Recall that $\Psi(z, \boldsymbol{x})$ represents a two-layer neural network with input $z$ and parameters $\boldsymbol{x}$. Assumption 3.1 holds when $\sup_z \sup_{\boldsymbol{x}} |\Psi(z,\boldsymbol{x})|$, $\sup_z \sup_{\boldsymbol{x}} \|\nabla_2 \Psi(z, \boldsymbol{x})\|$, $\sup_z \sup_{\boldsymbol{x}} \|\mathbf{H}_2 \Psi(z, \boldsymbol{x})\|_{\mathrm{op}}$ are bounded and when the loss function $\ell : \mathbb{R} \times \mathbb{R} \to \mathbb{R}$ is convex and Lipschitz with respect to its first argument [79]. These conditions on $\Psi$ are satisfied by smoothly-clipped two-layer neural networks with ReLU, tanh, or sigmoid activation functions. The conditions on $\ell$ are satisfied by both the squared loss and cross entropy loss. Let $\{(z_s, y_s)\}_{s=1}^n$ be all the training data*

*at each iteration. Assumption 3.2 holds with a bounded kernel $k(\boldsymbol{x},\boldsymbol{x}^\circ) = \sum_{s=1}^n \Psi(z_s, \boldsymbol{x})\Psi(z_s,\boldsymbol{x}^\circ)^2$ and its associated RKHS $\mathcal{H}_k$, in which $\|\Psi(z_s,\cdot)\|_{\mathcal{H}_k} \le 1$ for any $s \in \{1, \ldots, n\}$ (see Lemma B.10).*
*2. Quantization with MMD: Recall that $q_2$ is the kernel $\varkappa : \mathbb{R}^d \times \mathbb{R}^d \to \mathbb{R}$ used in MMD. Assumption 3.1 holds when $\varkappa$ is bounded (i.e. $\sup_{\boldsymbol{x},\boldsymbol{x}'} |\varkappa(\boldsymbol{x},\boldsymbol{x}')| < \infty$) and its first and second order derivatives are bounded (i.e. $\sup_{\boldsymbol{x},\boldsymbol{x}'} \|\nabla_1 \varkappa(\boldsymbol{x},\boldsymbol{x}')\| < \infty$, $\sup_{\boldsymbol{x},\boldsymbol{x}'} \|\mathbf{H}_1 \varkappa(\boldsymbol{x},\boldsymbol{x}')\|_{\mathrm{op}} < \infty$). Assumption 3.2 holds trivially with kernel $k = \varkappa$. These conditions include popular kernels like the Gaussian and Matérn of order at least $\frac{5}{2}$.*
*3. PrO Posteriors: Recall that $q_2(\boldsymbol{x},\boldsymbol{x}^\circ) = \iint \varkappa(y,y')p_{\boldsymbol{x}}(y)p_{\boldsymbol{x}^\circ}(y')\mathrm{d}y\mathrm{d}y'$ where $p_{\boldsymbol{x}}$ is the density for the statistical model $P_{\boldsymbol{x}}$. Assumption 3.1 holds when the kernel $\varkappa$ satisfies $\sup_{y,y'} |\varkappa(y,y')| < \infty$, and when $\sup_{\boldsymbol{x}} \int \|\nabla_{\boldsymbol{x}} p_{\boldsymbol{x}}(y)\|\mathrm{d}y < \infty$ and $\sup_{\boldsymbol{x}} \int \|\mathbf{H}_{\boldsymbol{x}} p_{\boldsymbol{x}}(y)\|_{\mathrm{op}}\mathrm{d}y < \infty$. Assumption 3.2 holds when $\sup_y \|\boldsymbol{x} \mapsto p_{\boldsymbol{x}}(y)\|_{\mathcal{H}_k} < \infty$ where $\mathcal{H}_k$ is an RKHS associated with a bounded kernel $k$ (cf. Lemma B.9). These conditions are satisfied, for example, when the kernel $k(x,x') = \exp(-\|m(x) - m(x')\|^2/(2\varsigma^2))$ and $P_{\boldsymbol{x}}(y) = \mathcal{N}(y \mid m(\boldsymbol{x}),\varsigma^2)$ with variance $\varsigma > 0$ and mean function $m : \mathbb{R}^d \to \mathbb{R}$ satisfies $\sup_{\boldsymbol{x}} \|\nabla m(\boldsymbol{x})\| < \infty$ and $\sup_{\boldsymbol{x}} \|\mathbf{H}m(\boldsymbol{x})\|_{\mathrm{op}} < \infty$ [cf. Theorem 2 in 57].*

## 4. Experiments

In this section, we provide empirical evidence supporting the claimed advantages of KT-MFLD in terms of the trade-off between computational cost and accuracy. In particular, we demonstrate that, when compared under the *same computational cost*, KT-MFLD consistently outperforms both the original MFLD and other subsampling-based methods, in agreement with the convergence guarantees established in Theorem 3.3 and associated computational complexities. In our experiments, we implement two vari-

---

[2]While the kernel is bounded for a fixed choice of $n$, the bound would increase with a larger $n$ which would deteriorate the quality of the samples from KT-SPLIT-Compress.

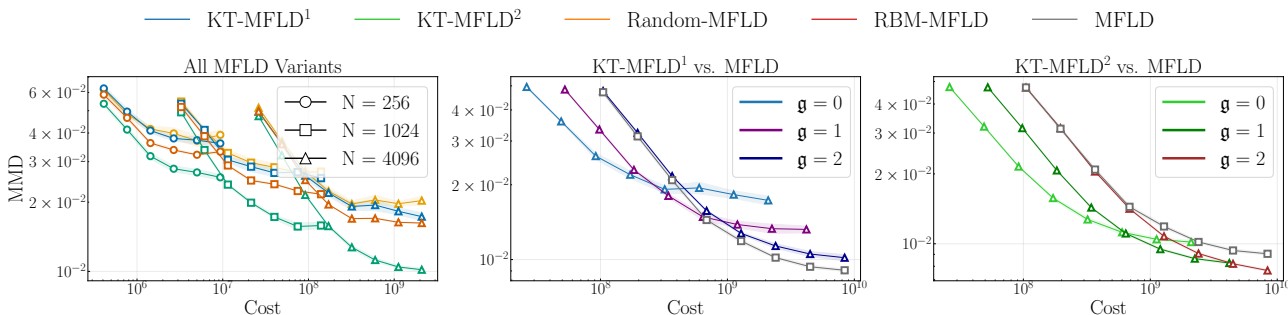

*Figure 3. MMD quantization.* **Left:** Comparison of `KT-MFLD`[1] and `KT-MFLD`[2] with all subsampling based acceleration methods. **Middle:** Comparison of `KT-MFLD`[1] with original `MFLD` under varying 𝔤. **Right:** Comparison of `KT-MFLD`[2] with original `MFLD` under varying 𝔤. Results are averaged over 20 independent runs. Shaded areas represent ±2 standard errors.

ants of `KT-MFLD`, differing only in the thinning procedure applied at each iteration: one using `KT-SPLIT`-Compress and the other using KT-Compress. We refer to these two variants as `KT-MFLD`[1] and `KT-MFLD`[2], respectively. The baselines we consider are standard mean-field Langevin dynamics with no thinning (`MFLD`), MFLD with random thinning (`Random-MFLD`), and the random batch method (`RBM-MFLD`). The first three experiments cover all the three illustrative applications of MFLD considered in Section 3 where all assumptions can hold (Remark 3.4); while the last mean-field game setting represents a closely-related application of particle simulations beyond the scope of MFLD. More experimental details can be found in Appendix C. The code to reproduce all experiments is available at https://github.com/hudsonchen/thinned_mfld.

**Mean Field Neural Networks** First, we consider the problem of training a mean-field neural network in a student–teacher setting. Given a teacher network $h_{\text{teacher}}(z) = \frac{1}{N_0} \sum_{j=1}^{N_0} \Psi(z, \boldsymbol{w}^{(j)})$, the dataset $\{z_i, y_i\}_{i=1}^{n}$ of size $n$ is generated with additive Gaussian noise corruption: $y_i = h_{\text{teacher}}(z_i) + \varsigma_i$ with $\varsigma_i \sim \mathcal{N}(0, 0.1)$. The teacher network parameters $\{\boldsymbol{w}^{(j)}\}_{j=1}^{N_0}$ are sampled from a mixture of 10 Gaussian distributions on $\mathbb{R}^d$ with $d = 12$ and are then held fixed throughout training. The covariates $\{z_i\}_{i=1}^{n}$ are sampled i.i.d. from a uniform distribution over the $d$-dimensional unit sphere. The student network shares the same functional form as the teacher, $h_{\text{student}}(z) = \sum_{j=1}^{N} \frac{1}{N} \Psi(z, \boldsymbol{x}^{(j)})$, whose parameters $\{\boldsymbol{x}^{(j)}\}_{j=1}^{N}$ are initialized randomly and trained via *noisy* gradient descent over the regularized empirical squared loss. During training, each update of a student particle uses a freshly generated batch of samples, corresponding to an online teacher–student learning setting [71, 5]. With $\ell_2$-regularization of strength $\zeta = 10^{-4}$ and additive noise of scale $\sigma = 10^{-3}$ injected at each iteration, the training procedure corresponds to MFLD simulated with $N$ particles, as outlined earlier.

In Figure 2, we report the performances of all methods evaluated by the loss on the test dataset holding against *equal total computational cost*, defined as Cost $:= N^\beta \times T$

where $\beta = 2$ for the original `MFLD` and $\beta = \frac{3}{2}$ for `KT-MFLD`, `Random-MFLD`, and `RBM-MFLD`. Although Assumption 3.2 could in principle be satisfied by using an empirical kernel over the full dataset (Remark 3.4), doing so would be computationally prohibitive. For this reason, we instead use a Sobolev kernel for both versions of `KT-MFLD` in our experiments. In the **Left** panel, we observe that `KT-MFLD`[1] and `KT-MFLD`[2] consistently attain smaller test loss than both `Random-MFLD` and `RBM-MFLD`. Given that these methods have the same cost per iteration, this amounts to showing that `KT-MFLD` consistently outperforms `Random-MFLD` and `RBM-MFLD` under the same number of particles $N$ and iterations $T$. In the **Middle** and **Right** panel, we compare the performance of `KT-MFLD`[1] and `KT-MFLD`[2] against the original `MFLD` under varying 𝔤 $= \{0, 1, 2\}$, where the cost scales as Cost $= 2^{\mathfrak{g}} \times N^{\frac{3}{2}} \times T$. Both variants of `KT-MFLD` achieve lower test loss than `MFLD` for all values of 𝔤, despite operating under a smaller computational budget while using a larger number of particles. Moreover, increasing 𝔤 consistently improves the final performance of both versions of `KT-MFLD` mitigating the extra logarithmic factor in $N$.

**Quantization with MMD** Now we consider the experiment of distribution quantization with MMD under the objective $F_0(\mu; \nu) = \iint \varkappa(\boldsymbol{x}, \boldsymbol{x}^\circ) \mathrm{d}\mu(\boldsymbol{x}) \mathrm{d}\mu(\boldsymbol{x}^\circ) - 2\varkappa(\boldsymbol{x}, \boldsymbol{y}) \mathrm{d}\mu(\boldsymbol{x}) \mathrm{d}\nu(\boldsymbol{y}) + \text{const}$, where the kernel $\varkappa$ is a Gaussian kernel with a fixed length scale 1. We follow the experimental setup of [22, Section 5.1], where $\nu$ is chosen to be a mixture of Gaussian distributions and the goal is to find a good approximation of $\nu$ (in the sense of MMD) using a finite number of particles. While the original formulation of [22] minimizes $F_0$ directly without either $\ell_2$ regularization or entropy regularization, we incorporate both here in this setting as a benchmark for MFLD and related thinning-based acceleration techniques. Specifically, we pick $\zeta = 10^{-4}$ and $\sigma = 10^{-3}$. We use the same step size $\gamma = 1.0$ for all the methods for a consistent comparison.

In Figure 3, we report the performance of all methods using the MMD computed with the same kernel $\varkappa$ as in the defini-

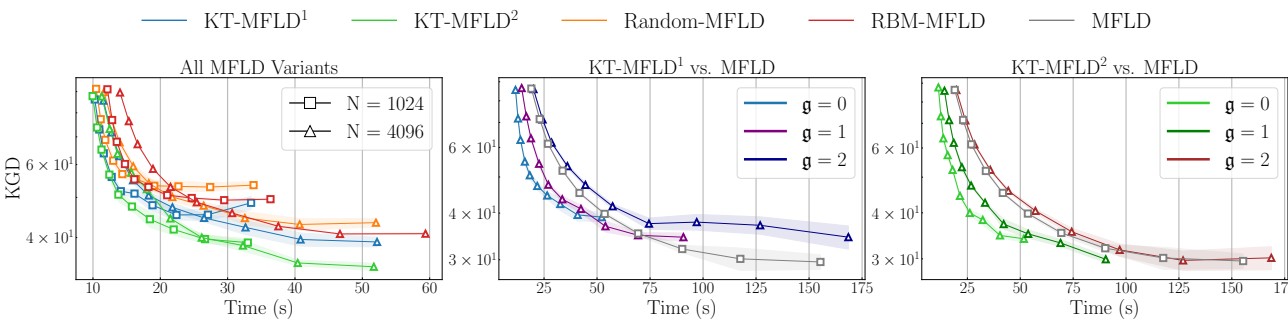

*Figure 4. PrO Posteriors.* **Left:** Comparison of `KT-MFLD` with all subsampling based acceleration methods. **Middle:** Comparison of `KT-MFLD`[1] with original `MFLD` under varying $\mathfrak{g}$. **Right:** Comparison of `KT-MFLD`[2] with original `MFLD` under varying $\mathfrak{g}$. Results are averaged over 20 independent runs. Shaded areas represent $\pm 2$ standard errors. The reported wall-clock time is measured on a AMD Ryzen 9 5950X CPU workstation with 16 physical cores and 32 threads.

tion of the objective $F_0$, under an *equal total computational cost*. We also use this kernel for both versions of `KT-MFLD` in our experiments. In the **Left** panel, we observe that `KT-MFLD`[1] and `KT-MFLD`[2] consistently produce smaller test loss than both `Random-MFLD` and `RBM-MFLD`. Given that these methods have the same cost per iteration, this again shows that `KT-MFLD` outperforms `Random-MFLD` and `RBM-MFLD` under the same number of particles $N$ and iterations $T$. In the **Middle** and **Right** panel, we compare the performance of `KT-MFLD`[1] and `KT-MFLD`[2] against the original `MFLD` under varying $\mathfrak{g} = \{0, 1, 2\}$. In this experiment, `KT-MFLD`[1] is outperformed by `MFLD`, whereas `KT-MFLD`[2] consistently attains a lower MMD than `MFLD` across all values of $\mathfrak{g}$. The superior performance of `KT-MFLD`[2] over `KT-MFLD`[1] can be attributed to the strong MMD guarantee of KT-Compress [76], which is particularly well-suited in this task of MMD minimization.

**PrO Posteriors** We consider parameter inference for a misspecified Lotka–Volterra model following [74, Section 6.1]. In this set-up, the statistician assumes the prey ($u_1$) and the predator population ($u_2$) follow a Lotka–Volterra model (LVM), which captures the cyclical rise and fall driven by predation and reproduction [83]. The dynamics are governed by the coupled differential equations

$$
\begin{aligned}
\mathrm{d}u_1/\mathrm{d}\tau &= \mathfrak{a}u_1 - \mathfrak{b}u_1u_2, \quad u_1(0) = \xi_1, \\
\mathrm{d}u_2/\mathrm{d}\tau &= \mathfrak{c}u_1u_2 - \mathfrak{d}u_2, \quad u_2(0) = \xi_2,
\end{aligned} \tag{10}
$$

with non-negative parameters $\boldsymbol{x} = (\mathfrak{a}, \mathfrak{b}, \mathfrak{c}, \mathfrak{d})$ and given initial conditions $\xi_1, \xi_2$. However, the actual trajectories $\boldsymbol{u}(\tau) = [u_1(\tau), u_2(\tau)]$ arise from a *stochastic* LVM, meaning that Eq. (10) is misspecified, a setting where standard Bayesian inference can fail.

Observations $y_i$ of both species are made at discrete times $\tau_i$, corrupted by independent Gaussian noise of variance 1. The statistician's model $P_{\boldsymbol{x}}$ has density

$$
p_{\boldsymbol{x}}(y_i \mid \tau_i) = \prod_i \frac{1}{\sqrt{2\pi}} \exp\left(-\frac{\|y_i - \boldsymbol{u}_{\boldsymbol{x}}(\tau_i)\|^2}{2}\right),
$$

where $\boldsymbol{u}_{\boldsymbol{x}}(\tau_i)$ represents the outcome of the LVM Eq. (10) simulated until time $\tau_i$. All simulation settings were identical to those used in [74]. Aware of the potential for model misspecification, the statistician eschews the Bayesian posterior in favour of the PrO posterior based on the kernel scoring rule, defined as the minimizer of an entropy-regularized objective in Eq. (1) with

$$
F_0(\mu) = \mathrm{MMD}^2\left(\int \prod_i p_{\boldsymbol{x}}(\cdot \mid \tau_i)\mathrm{d}\mu(\boldsymbol{x}), \prod_i \delta_{y_i}\right), \tag{11}
$$

where the MMD here is computed under a product of Gaussian kernels with fixed length scale 1 (i.e. for each time point $\tau_i$) was used. Our task is now to numerically approximate the PrO posterior defined by the minimizer of the objective $\mathscr{F}(\mu) := F_0(\mu) + \frac{\zeta}{2}\mathbb{E}_\mu[\|\boldsymbol{x}\|^2] - \sigma\mathrm{Ent}(\mu)$. Here, we take $\zeta = 0.1$ and $\sigma = 10^{-3}$.

In Figure 4, we report the performances of all the methods under an evaluation metric called *kernel gradient discrepancy* (KGD), a divergence specifically designed to measure discrepancies with respect to distributions $\pi$ defined implicitly as minimizers of functionals in Eq. (1) [16]. We use Sobolev kernels for both version of `KT-MFLD`. In Figure 4, the comparison is done under same computational *time* in wall-clock. In the **Left** panel, both `KT-MFLD`[1] and `KT-MFLD`[2] consistently achieve lower KGD than `Random-MFLD` and `RBM-MFLD`. In the **Middle** and **Right** panels, we observe that both variants of `KT-MFLD` attain smaller KGD than `MFLD` for $\mathfrak{g} = 0$ under the same computational time, despite operating with a larger number of particles. In this setting, evaluating the interaction between any pair of particles $\boldsymbol{x}^{(i)}$ and $\boldsymbol{x}^{(j)}$ requires solving Eq. (10), which constitutes the dominant computational cost. As a result, the superior performance of `KT-MFLD` at a fixed computational budget directly translates into superior performance under equal wall-clock time.

**Mean-field Games** Beyond MFLD, a closely related framework is that of *mean-field games* (MFGs), which characterize equilibrium configurations of noncooperative differential

games with a continuum of interacting agents [52, 7]—a topic receiving increasing interest in multi-agent reinforcement learning [87] and generative modeling [8, 55]. Numerical simulation of MFGs similarly incurs an $\Omega(N^2)$ computational cost per time step, due to the evaluation of pairwise interactions among $N$ agents. We hope to use the same thinning technique to reduce this computational cost. Following [1], we consider a MFG in which each agent chooses a trajectory so as to minimize a risk functional that balances control effort and interaction with the population. Specifically, at equilibrium the value function $\phi : [0, T] \times \mathbb{R}^d \to \mathbb{R}$ and distribution $\rho_t$ satisfy the coupled Hamilton–Jacobi–Bellman and continuity equations:

$$-\partial_t \phi(t, \boldsymbol{x}) + \frac{1}{2}|\nabla\phi(t, \boldsymbol{x})|^2 = \int K(\boldsymbol{x}, \boldsymbol{y})\mathrm{d}\rho_t(\boldsymbol{y}),$$
$$\partial_t \rho_t(\boldsymbol{x}) - \nabla \cdot \big(\rho_t(\boldsymbol{x})\nabla\phi(t, \boldsymbol{x})\big) = 0, \qquad (12)$$

subject to the initial condition $\rho_0 = \nu$ and terminal condition $\phi(T, \boldsymbol{x}) = \psi(\boldsymbol{x})$. The trajectories induced by the coupled system in Eq. (12), minimize the risk functional $J(z) := \int_0^T [\frac{1}{2}|\frac{\mathrm{d}}{\mathrm{d}s}z(s)|^2 - \int K\big(z(s), \boldsymbol{y}\big)\mathrm{d}\rho_s(\boldsymbol{y})]\mathrm{d}s + \psi\big(z(T)\big)$ over all admissible trajectories $z : [0, T] \to \mathbb{R}^d$.

We numerically solve Eq. (12) using $N = 4096$ particles over the time interval $[0, 1]$ with step size $\Delta t = 0.01$. The terminal cost function is $\psi(\boldsymbol{x}) = 10\|\boldsymbol{x} - \boldsymbol{x}_{\text{target}}\|^2$ with $\boldsymbol{x}_{\text{target}} = 0$. The kernel interaction term $K$ is a Gaussian kernel with length scale $1.0$. The particles are initialized by sampling from a mixture of eight Gaussian distributions $\rho_0$. We refer to the basic method as MFG. We additionally solve Eq. (12) using thinning-based accelerations, where the kernel interaction for each particle is evaluated on a thinned subset selected either by random subsampling (Random-MFG), KT-SPLIT-Compress (KT-MFG$^1$), KT-Compress (KT-MFG$^2$). The kernel used for both KT-MFG is a Gaussian kernel with length scale $1.0$. The random batch method [44] is not applicable in this setting. In Figure 5, we observe that both KT-MFLD$^1$ and KT-MFLD$^2$ consistently achieve lower risk than Random-MFLD and RBM-MFLD under the same computational wall-clock-time. This proves the applicability of KT-SPLIT-Compress and KT-Compress in reducing the cost of simulating particles beyond MFLD.

## 5. Conclusions, Limitations & Future Work

Several important tasks in machine learning can be formulated as the minimization of entropy-regularized functionals, for which computational efficiency is often a major bottleneck. Our proposed KT-MFLD provides a step toward alleviating this challenge. A key limitation of the present work is the relatively modest scale of our experiments. Since this is primarily a theoretical paper in the context of MFLD, it is difficult to identify large-scale experimental settings

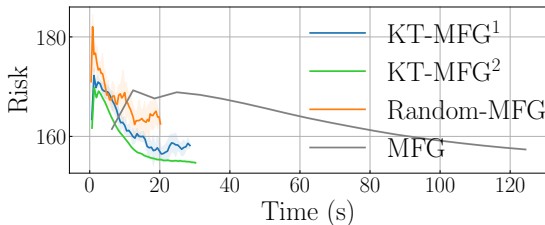

*Figure 5. Mean-field Games.* Results are averaged over 20 independent runs. Shaded areas represent $\pm 2$ standard errors.

that both satisfy our assumptions and are natural benchmarks; consequently, our experiments are mainly restricted to kernel-based particle flows. Promising future directions include combining KT-MFLD with momentum-based acceleration techniques [54] and applying kernel thinning to particle simulations beyond MFLD and MFG, such as Stein variational gradient descent [56].

## Acknowledgement

ZC was supported by the Engineering and Physical Sciences Research Council (EPSRC) EP/S021566/1. HK and CJO were supported by EP/W019590/1. CJO was supported by a Philip Leverhulme Prize PLP-2023-004. FXB was supported by EPSRC EP/Y022300/1. ZC would like to thank Zhichao Chen for helpful discussion on the implementation of mean field games and for pointing out relevant references.

## Impact Statement

This paper presents work whose goal is to advance the field of Machine Learning. There are many potential societal consequences of our work, none which we feel must be specifically highlighted here.

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

# Supplementary Material

This appendix is structured as follows. In Appendix A, we first discuss how thinning could enhance the performance of general interacting particle systems beyond MFLD, then in Appendix B we provide proofs of our theoretical results. Finally, in Appendix C we provide additional experimental details for Section 4.

## A. Thinned Interacting Particle Systems

Our paper showed that thinning can be provably beneficial in the context of MFLD, reducing the computational complexity whilst maintaining control of the bias. In the following section, we discuss other interacting particle systems which could benefit from this approach and could warrant further work.

**Thinned McKean–Vlasov Diffusion Processes** A McKean–Vlasov diffusion process $(X_t)_{t\geq 0}$ is defined as a stochastic process whose dynamics depend on its own law. Formally, given measurable functions $G : \mathbb{R}^d \times \mathcal{P}(\mathbb{R}^d) \to \mathbb{R}^d, \quad \Lambda : \mathbb{R}^d \times \mathcal{P}(\mathbb{R}^d) \to \mathbb{R}^d$, a process $(X_t)_{t\geq 0}$ is called a McKean–Vlasov diffusion if it satisfies [28]:

$$\mathrm{d}X_t = G(X_t, \mu_t)\mathrm{d}t + \Lambda(X_t, \mu_t)\,\mathrm{d}B_t, \quad \mathrm{Law}(X_t) = \mu_t. \tag{A.1}$$

The existence and uniqueness of the solution can be guaranteed by standard Lipschitz and growth conditions on $G$ and $\Lambda$.

MFLD is a special case of McKean–Vlasov diffusion processes with $G(\boldsymbol{x}, \mu) = \nabla F_0'[\mu](\boldsymbol{x})$ and constant diffusion coefficient $\Lambda(\boldsymbol{x}, \mu) = \sqrt{2\sigma}$. For McKean–Vlasov diffusion processes, the drift and/or diffusion coefficients often depend on $\mu_t$ through integral operators, reflecting the fact that these equations describe the *mean-field limit* of large systems of interacting particles [13], i.e., $G(\boldsymbol{x}, \mu) = \int g(\boldsymbol{x}, \boldsymbol{y})\mathrm{d}\mu(\boldsymbol{y})$ and $\Lambda(\boldsymbol{x}, \mu) = \int L(\boldsymbol{x}, \boldsymbol{y})\mathrm{d}\mu(\boldsymbol{y})$ for two functions $g : \mathbb{R}^d \times \mathbb{R}^d \to \mathbb{R}$ and $L : \mathbb{R}^d \times \mathbb{R}^d \to \mathbb{R}$. From now on, we only consider McKean–Vlasov diffusion processes whose drift and diffusion term enjoy such forms. Such form of McKean–Vlasov diffusion processes already cover all the forms of MFLD considered in the main text, and also include wide applications of mean-field games [52, 7], inverse problems [27], missing data imputation [19] and computational finance [30].

In this case, a practical implementation of Eq. (A.1) under both time- and space- discretizations is the following interacting particle system: given an initial collection of particles $\mathscr{X}_0 = [\boldsymbol{x}_0^{(1)}, \dots, \boldsymbol{x}_0^{(N)}]$ and a time horizon $T \in \mathbb{N}$, for each time $t \in \{0, \dots, T\}$ and for each particle $i \in \{1, \dots, N\}$,

$$\boldsymbol{x}_{t+1}^{(i)} = \boldsymbol{x}_t^{(i)} - \gamma G(\boldsymbol{x}_t^{(i)}, \mu_{\mathscr{X}_t}) + \sqrt{2\gamma \Lambda(\boldsymbol{x}_t^{(i)}, \mu_{\mathscr{X}_t})}\, \xi_t^{(i)}, \quad \mu_{\mathscr{X}_t} = \frac{1}{N}\sum_{i=1}^N \delta_{\boldsymbol{x}_t^{(i)}}, \quad \xi_t^{(i)} \sim \mathcal{N}(0, \mathrm{Id}). \tag{A.2}$$

Here, both $G(\boldsymbol{x}_t^{(i)}, \mu_{\mathscr{X}_t}) = \frac{1}{N}\sum_{j=1}^N g(\boldsymbol{x}_t^{(i)}, \boldsymbol{x}_t^{(j)})$ and $\Lambda(\boldsymbol{x}_t^{(i)}, \mu_{\mathscr{X}_t}) = \frac{1}{N}\sum_{j=1}^N L(\boldsymbol{x}_t^{(i)}, \boldsymbol{x}_t^{(j)})$ involve an empirical average over $N$ particles, so that the implementation of the above descent scheme requires the cost of $\mathcal{O}(N^2)$ at each iteration. The thinning techniques KT-SPLIT-Compress and KT-Compress apply equally to this more general McKean–Vlasov descent scheme, in which it selects a coreset of size $M = \lceil\sqrt{N}\rceil$ and thus reduces the cost to $\mathcal{O}(MN) = \mathcal{O}(N^{\frac{3}{2}})$ per iteration. However, theoretical analysis for convergence of McKean–Vlasov diffusion processes with both space- and time-discretizations are less developed in the literature than those of mean-field Langevin dynamics (MFLD), as the former no longer corresponds to a descent scheme that minimizes a specific functional. Consequently, the McKean–Vlasov literature has primarily focused on convergence of particle-based discretizations to the population-level limit, rather than convergence towards the minimizer; see, for example, [2].

**Thinned MFLD of General Functionals** Another direction to generalize Eq. (2) is to consider more general objectives $F : \mathcal{P}_2(\mathbb{R}^d) \to \mathbb{R}$. In the main text, we focus on a class of non-linear functionals $F$ of the form $F(\mu) = F_0(\mu) + \frac{\varsigma}{2}\mathbb{E}_\mu[\|\boldsymbol{x}\|^2]$, where

$$F_0(\mu) = \mathbb{E}_{u\sim\rho}\left[R_1\left(\int q_1(u, \boldsymbol{x})\mathrm{d}\mu(\boldsymbol{x})\right)\right] + \frac{1}{2}\iint q_2(\boldsymbol{x}, \boldsymbol{x}^\circ)\mathrm{d}\mu(\boldsymbol{x})\mathrm{d}\mu(\boldsymbol{x}^\circ). \tag{A.3}$$

The functional $F_0$ considered above involves interactions of order at most two. More generally, it can be extended to the following higher-order form:

$$F_0(\mu) = \sum_{i=1}^{p} \mathbb{E}_{u \sim \rho} \left[ R_i \left( \int q_i(u, \boldsymbol{x}_1, \ldots, \boldsymbol{x}_i) \mathrm{d}\mu^{\otimes i}(\boldsymbol{x}) \right) \right], \tag{A.4}$$

which recovers the original definition when $p = 2$ and $R_2$ is the identity map. For this general class of functionals, the associated vector field in the particle update scheme requires approximating nested integrals of order $p$, which incurs a computational cost of $\mathcal{O}(N^p)$ at each iteration. By applying the same thinning strategy, KT-SPLIT-Compress or KT-Compress used in the main text, this cost can be reduced to $\mathcal{O}(N^{\frac{p+1}{2}})$. Establishing convergence guarantees analogous to those in Theorem 3.3 for MFLD of such functionals would, however, require more sophisticated theoretical analysis, due to the increased complexity induced by higher-order particle interactions. Such form of $F_0$ with higher-order interactions arise prominently in statistical physics [37].

**Thinned Stein Variational Gradient Descent**    Stein variational gradient descent (SVGD) concerns the computational task of sampling from an un-normalized target distribution $\pi \propto \exp(-V)$. Its update scheme can be written as follows [56]: for initial particles $\mathscr{X}_0 = [\boldsymbol{x}_0^{(1)}, \ldots, \boldsymbol{x}_0^{(N)}]$ and $t \in \{0, \ldots, T\}$ for any $T \in \mathbb{N}$, for $i = 1, \ldots, N$,

$$\boldsymbol{x}_t^{(i+1)} = \boldsymbol{x}_t^{(i)} - \gamma \frac{1}{N} \sum_{j=1}^{N} \left[ k(\boldsymbol{x}_t^{(i)}, \boldsymbol{x}_t^{(j)}) \nabla V(\boldsymbol{x}_t^{(j)}) - \nabla_2 k(\boldsymbol{x}_t^{(i)}, \boldsymbol{x}_t^{(j)}) \right].$$

The particle interaction term plays a crucial role by inducing repulsion among the $N$ particles, thereby promoting sample diversity and mitigating particle degeneracy. However, this benefit comes at the cost of increased computational complexity: the interaction term induces a per-iteration cost of $\mathcal{O}(N^2)$ due to pairwise interactions.

Unlike the MFLD setting considered in the main text, the coreset produced by KT-SPLIT-Compress does not in general achieve the same $\mathcal{O}(N^{-\frac{1}{2}})$ for the vector field after thinning as established for MFLD in Proposition B.1. The underlying reason is that the mapping $\boldsymbol{x} \mapsto k(\boldsymbol{x}^{(i)}, \boldsymbol{x}) \nabla V(\boldsymbol{x})$ does not belong to the RKHS $\mathcal{H}_k$ associated with $k$ due to multiplication with $\nabla V$. Moreover, this mapping may fail to lie in any *bounded* RKHS, since $\nabla V$ is typically not bounded (for instance, in the Gaussian case one has $\nabla V(\boldsymbol{x}) \propto \boldsymbol{x}$). The same issue exists for other variants of SVGD as well including *nonlinear Stein variational gradient descent* [82] and *kernel gradient descent* [16]. Therefore, applying thinning to SVGD and its related variants may be more challenging than for the other interacting particle systems discussed above.

**Other Interacting Particle Systems**    Beyond the MFLD setting studied in the main text and the generalizations discussed above, there exist many other applications in which simulating interacting particles incurs at least quadratic computational cost per iteration. Representative examples include mean field games [7], ensemble Kalman filters [47], consensus-based optimization [15], interacting Markov chain Monte Carlo methods [70], and sequential Monte Carlo [31]. Understanding whether and how kernel thinning can be adapted to these settings is an interesting direction for future work.

**Other Acceleration Techniques**    Beyond particle subsampling to reduce the per-iteration cost of evaluating interactions like KT-MFLD, a variety of alternative acceleration strategies have been proposed for interacting particle systems. These include Nesterov-type accelerated gradient methods in the space of probability measures [54, 77], regularization of kernel integral operators [41, 20], and random Fourier features to approximate kernel interactions [12, 1]. In addition, in the context of mean-field Langevin dynamics for training two-layer neural networks, the cost of computing the vector field can be reduced by subsampling the training data [79, 84]. Importantly, these methods are all complementary to KT-MFLD: for instance, data subsampling can be combined with both KT-SPLIT-Compress and KT-Compress, and same for momentum-based acceleration as well. Understanding how these acceleration strategies interact, and whether they can be combined to yield further computational gains, is therefore an interesting direction for future work.

## B. Proofs

In this appendix, we provide the proofs of all theoretical results in the paper and also present a number of intermediate results. More precisely, we give the proof of Proposition 2.1 in Appendix B.1, the proof of Theorem 3.3 in Appendix B.2, and the proof of auxiliary propositions and lemmas in Appendix B.3, Appendix B.4, and Appendix B.5.

**Additional Notation** We recall that $\mathcal{H}_k^{\otimes d}$ denotes the Cartesian product RKHS consisting of elements $f = (f_1, \ldots, f_d)$ with $f_i \in \mathcal{H}_k$ and with inner product $\langle f, g \rangle_{\mathcal{H}_k^{\otimes d}} = \sum_{i=1}^d \langle f_i, g_i \rangle_{\mathcal{H}_k}$.

### B.1. Proof of Proposition 2.1

*Proof.* The proof is a direct application of Theorem 3 of [76] combined with its Example 3. Given the input points $\{x_i\}_{i=1}^N$ with size $N$, consider using the KT-SPLIT-Compress($\delta$) algorithm with a bounded kernel $k$ (i.e. $\|k\|_\infty := \sup_x k(x, x) < \infty$.) and a hyperparameter $\mathfrak{g} \in \mathbb{N}_0$ which returns a thinned coreset $\{\bar{x}_i\}_{i=1}^M$ of size $M = 2^\mathfrak{g} \sqrt{N}$.

From Example 3 of [76], the thinned coreset satisfies the following: with probability at least $1 - \frac{\delta}{2}$, for any $f \in \mathcal{H}_k$,

$$\left| \frac{1}{N} \sum_{i=1}^N f\left(x^{(i)}\right) - \frac{1}{M} \sum_{i=1}^M f\left(\bar{x}^{(i)}\right) \right| \leq \|f\|_{\mathcal{H}_k} \cdot \nu_0 \sqrt{2 \log\left(\frac{2}{\delta}\right)}, \tag{B.1}$$

where, with probability $1 - \frac{\delta}{2}$,

$$\nu_0 \leq \sqrt{\log_4 N - \mathfrak{g}} \frac{4}{2^{\mathfrak{g}+1} \sqrt{N} \sqrt{3}} \sqrt{\log\left(\frac{12 N 4^\mathfrak{g} (\log_4 N - \mathfrak{g})}{2^{\mathfrak{g}+1} \sqrt{N} \delta}\right) \|k\|_\infty} \tag{B.2}$$

$$= \frac{2}{M\sqrt{3}} \sqrt{\log_2\left(\frac{N}{M}\right) \log\left(\frac{6M}{\delta} \log_2\left(\frac{N}{M}\right)\right) \|k\|_\infty}. \tag{B.3}$$

The first probability statement is about the randomness of the thinned coreset, while the second probability statement is about a specific intrinsic thresholding step in the KT-split algorithm (Algorithm 3 in [35]). This concludes the proof of Proposition 2.1.

Next, we consider $\mathfrak{g} = 0$ as a special case such that $M = \sqrt{N}$. By the union bound, we therefore obtain that, with probability at least $1 - \delta$,

$$\left| \frac{1}{N} \sum_{i=1}^N f\left(x^{(i)}\right) - \frac{1}{M} \sum_{i=1}^M f\left(\bar{x}^{(i)}\right) \right| \leq \|f\|_{\mathcal{H}_k} \cdot \sqrt{2\log\left(\frac{2}{\delta}\right)} \cdot \sqrt{\log_4 N} \frac{2}{\sqrt{N}\sqrt{3}} \sqrt{\log\left(\frac{6\sqrt{N}(\log_4 N)}{\delta}\right) \|k\|_\infty}$$

$$\leq \frac{3\sqrt{\log N}}{\sqrt{N}} \|f\|_{\mathcal{H}_k} \sqrt{\|k\|_\infty} \cdot \sqrt{\log\left(\frac{2}{\delta}\right)\left(\log\left(\frac{1}{\delta}\right) + \log N\right)}.$$

To remove the extra logarithmic term in the above upper bound, we consider $\mathfrak{g} = \lceil \log_2 \log N \rceil$. We have, for $N$ sufficiently large that $(\log N)^2 \leq \sqrt{N}\delta$, with probability at least $1 - \delta$,

$$\left| \frac{1}{N} \sum_{i=1}^N f\left(x^{(i)}\right) - \frac{1}{M} \sum_{i=1}^M f\left(\bar{x}^{(i)}\right) \right|$$

$$\leq \|f\|_{\mathcal{H}_k} \cdot \sqrt{2\log\left(\frac{2}{\delta}\right)} \cdot \sqrt{\log_4 N} \frac{2}{(\log N)\sqrt{N}\sqrt{3}} \sqrt{\log(6\sqrt{N}(\log N)(\log_4 N)) \|k\|_\infty}$$

$$\leq \frac{6}{\sqrt{N}} \|f\|_{\mathcal{H}_k} \sqrt{\|k\|_\infty} \cdot \sqrt{\log\left(\frac{2}{\delta}\right)}.$$

Therefore, with such choice of $\mathfrak{g}$, we can further avoid the logarithmic term in $N$, achieving $\mathcal{O}(N^{-\frac{1}{2}})$ integration error. $\qquad\square$

### B.2. Proof of Theorem 3.3

To simplify the statement of the proof, we introduce the following shorthand notation:

$$v\left(x_t^{(i)}; \left\{x_t^{(j)}\right\}_{j=1}^N\right) := \nabla F_0'[\mu_{\mathscr{X}_t}]\left(x_t^{(i)}\right) \quad \text{and} \tag{B.4}$$

$$\omega\left(\boldsymbol{x}_t^{(i)}; \left\{\boldsymbol{x}_t^{(j)}\right\}_{j=1}^N, \left\{\bar{\boldsymbol{x}}_t^{(j)}\right\}_{j=1}^M\right) := \nabla F_0'[\mu_{\bar{\mathscr{X}}_t}]\left(\boldsymbol{x}_t^{(i)}\right) - \nabla F_0'[\mu_{\mathscr{X}_t}]\left(\boldsymbol{x}_t^{(i)}\right). \tag{B.5}$$

Here, $v$ denotes the velocity field of the original mean field Langevin dynamics, while $\omega$ denotes the additional approximation error induced by using a thinned coreset $\{\bar{\boldsymbol{x}}_t^{(i)}\}_{i=1}^M$ rather than the full set of particles $\{\boldsymbol{x}_t^{(i)}\}_{i=1}^N$ at each iteration. Now we write out the original mean field Langevin dynamics update scheme (Eq. (5)) and the thinned mean field Langevin dynamics update scheme (Eq. (7)): at iteration $t \in \mathbb{N}$, for $i = 1, \ldots, N$,

$$\boldsymbol{x}_{t+1}^{(i)} = \boldsymbol{x}_t^{(i)} - \gamma\left\{v\left(\boldsymbol{x}_t^{(i)}; \{\boldsymbol{x}_t^{(j)}\}_{j=1}^N\right) + \zeta\boldsymbol{x}_t^{(i)}\right\} + \sqrt{2\sigma\gamma}\,\xi_t^{(i)} \quad \text{and} \tag{MFLD}$$

$$\boldsymbol{x}_{t+1}^{(i)} = \boldsymbol{x}_t^{(i)} - \gamma\left\{v\left(\boldsymbol{x}_t^{(i)}; \{\boldsymbol{x}_t^{(j)}\}_{j=1}^N\right) + \omega\left(\boldsymbol{x}_t^{(i)}; \{\boldsymbol{x}_t^{(j)}\}_{j=1}^N, \{\bar{\boldsymbol{x}}_t^{(j)}\}_{j=1}^M\right) + \zeta\boldsymbol{x}_t^{(i)}\right\} + \sqrt{2\sigma\gamma}\,\xi_t^{(i)}. \tag{KT-MFLD}$$

Hereafter, we will omit the dependence on point sets $\{\bar{\boldsymbol{x}}_t^{(i)}\}_{i=1}^M$ and $\{\boldsymbol{x}_t^{(i)}\}_{i=1}^N$ for further notational simplicity. Our next result is proved in Appendix B.3.

**Proposition B.1** (Thinned vector field). *Suppose Assumptions 3.1 and 3.2 hold. Let $\nabla F_0'[\mu_\mathscr{X}] : \mathbb{R}^d \to \mathbb{R}^d$ be the original vector field evaluated at the empirical distribution of all $N$ particles $\mu_\mathscr{X} = \frac{1}{N}\sum_{i=1}^N \delta_{\boldsymbol{x}^{(i)}}$, and let $\nabla F_0'[\mu_{\bar{\mathscr{X}}}] : \mathbb{R}^d \to \mathbb{R}^d$ be the thinned vector field evaluated at the empirical distribution of the thinned set of $M$ particles $\mu_{\bar{\mathscr{X}}} = \frac{1}{M}\sum_{i=1}^M \delta_{\bar{\boldsymbol{x}}^{(i)}}$ with $M = \lceil\sqrt{N}\rceil$, obtained via* KT-SPLIT-*Compress($\delta$). Then, there exists a probabilistic event $\mathcal{E}(\delta)$ of probability at least $1 - \delta$, such that for any $\boldsymbol{x} \in \mathbb{R}^d$,*

$$\mathbb{E}\left[\|\nabla F_0'[\mu_\mathscr{X}](\boldsymbol{x}) - \nabla F_0'[\mu_{\bar{\mathscr{X}}}](\boldsymbol{x})\|^2 \cdot \mathbb{1}_{\mathcal{E}(\delta)}\right] \leq \frac{dC}{N}\log\left(\frac{6\sqrt{N}\log_4 N}{\delta}\right). \tag{B.6}$$

*Here, $C$ is a constant that only depends on $Q_1, L_R, \mathscr{Q}_1, \mathscr{Q}_2$.*

Since $C_1$ is a uniform upper bound on $\|\nabla F_0'[\mu](\boldsymbol{x})\|$ proved in Lemma B.3. From Proposition B.1, we can obtain that

$$\mathbb{E}\left[\left\|\omega\left(\boldsymbol{x}_t^{(i)}\right)\right\|^2\right] = \mathbb{E}\left[\|\nabla F_0'[\mu_\mathscr{X}](\boldsymbol{x}) - \nabla F_0'[\mu_{\bar{\mathscr{X}}}](\boldsymbol{x})\|^2\right] \leq \frac{dC}{N}\log\left(\frac{6\sqrt{N}\log_4 N}{\delta}\right) + C_1^2\delta.$$

We pick $\delta = (\log_2 N)^3/N$ and then obtain

$$\mathbb{E}\left[\left\|\omega\left(\boldsymbol{x}_t^{(i)}\right)\right\|^2\right] \leq \frac{dC\log(6N^{1.5}) + C_1^2(\log_2 N)^3}{N} \leq \frac{2dC\log N + 8C_1^2(\log N)^3}{N} =: \frac{c_0(\log N)^3}{N}. \tag{B.7}$$

Here, $C_0$ is a constant that only depends on $Q_1, L_R, \mathscr{Q}_1, \mathscr{Q}_2$. Now, we are ready to adapt the proof of [65] and [68] under our setting.

Recall that the objective functional $F(\mu)$ and $\mathscr{F}(\mu) = F_0(\mu) + \frac{\zeta}{2}\mathbb{E}_\mu[\|\boldsymbol{x}\|^2] - \sigma\mathrm{Ent}(\mu)$ defined in Eq. (3). Following [68], define a new auxiliary functional $\mathscr{F}^{(N)} : \mathcal{P}_2((\mathbb{R}^d)^N) \to \mathbb{R}$,

$$\mathscr{F}^{(N)}\left(\mu^{(N)}\right) = N\mathbb{E}_{\mathscr{X}\sim\mu^{(N)}}[F_0(\mu_\mathscr{X})] + \frac{\zeta}{2}\mathbb{E}_{\mathscr{X}\sim\mu^{(N)}}\left[\sum_{i=1}^N\left\|\boldsymbol{x}^{(i)}\right\|^2\right] - \sigma\mathrm{Ent}(\mu^{(N)}),$$

where $\mu^{(N)} \in \mathcal{P}_2((\mathbb{R}^d)^N)$ is the joint distribution of all $N$ particles $\mathscr{X} = [\boldsymbol{x}^{(1)}, \ldots, \boldsymbol{x}^{(N)}]$ and $\mu_\mathscr{X} = \frac{1}{N}\sum_{i=1}^N \delta_{\boldsymbol{x}^{(i)}}$ represents the empirical distribution. Note that the sign before entropy is flipped in our definition because the entropy in [68] is actually *negative* entropy. Denote $\pi = \arg\min_\mu \mathscr{F}(\mu)$ and $\pi^{\otimes N}$ is the N-fold tensor product of $\pi$.

**Proposition B.2.** *Suppose Assumptions 3.1 and 3.2 hold. Suppose $\gamma < \frac{1}{2\zeta}$. Let $\mu_t^{(N)}$ be the joint distribution of the $N$ particles in Eq. (7) at iteration $t \in \mathbb{N}$. Define $\bar{\alpha} = \frac{\zeta}{2\sigma}\exp(-\frac{4C_3}{\zeta\sigma}\sqrt{\frac{2d}{\pi}}))$ for $C_3 = (|R_1'(0)| + L_R Q_1 + Q_2)^2$. Then, for any $t \in \mathbb{N}$ we have,*

$$\mathbb{E}\left[\mathscr{F}^{(N)}(\mu_{t+1}^{(N)})\right] - N\mathscr{F}(\pi) - B - \frac{N\delta_\gamma + c_0(\log N)^3}{2\bar{\alpha}}$$

$$\leq \exp(-\bar{\alpha}\gamma\sigma)\left(\mathbb{E}\left[\mathscr{F}^{(N)}(\mu_t^{(N)})\right] - N\mathscr{F}(\pi) - B - \frac{N\delta_\gamma + c_0(\log N)^3}{2\bar{\alpha}}\right).$$

*Here, $\delta_\gamma = 8(C_2^2 + \zeta^2)(\gamma C_1^2 + \sigma d) + 32\gamma\zeta^2(C_2^2 + \zeta^2)(d + \frac{1}{\zeta}(\frac{C_1^2}{4\zeta} + \sigma d)) = \mathcal{O}(\gamma\sigma d + \gamma^2)$ represents the time discretization error, where $B = 2L_R Q_1^2 + Q_2$ is introduced in [Lemma B.6]{.underline}, $C_1 = |R_1'(0)| + L_R Q_1 + Q_2$ and $C_2 = Q_1^2 L_R + Q_2$ are introduced in [Lemma B.3]{.underline}, and $c_0$ is a constant defined in Eq. [(B.7)]{.underline}. The expectation is taken with respect to the randomness of the thinned samples at iteration $t$.*

The proof of the above proposition can be found in [Appendix B.4]{.underline}. $\bar{\alpha}$ is known as the *log Sobolev constant* in the literature of mean field Langevin dynamics [79, Lemma 5]. The proof is an adaptation of the proof of Theorem 2 in [65] in our setting where we take into account the extra error introduced by thinning at each iteration.

With [Proposition B.2]{.underline}, we can then obtain that, for any $T \in \mathbb{N}$,

$$\frac{1}{N}\mathbb{E}\left[\mathscr{F}^{(N)}(\mu_T^{(N)})\right] - \mathscr{F}(\pi) \leq \exp(-T\bar{\alpha}\gamma)\left(\frac{1}{N}\mathbb{E}\left[\mathscr{F}^{(N)}(\mu_0^{(N)})\right] - \mathscr{F}(\pi)\right) + \frac{B + c_0\bar{\alpha}^{-1}(\log N)^3}{N} + \frac{\delta_\gamma}{2\bar{\alpha}}.$$

Since the initial particles are sampled i.i.d from some distribution $\mu_0$, we have $\mu_0^{(N)} = \mu_0^{\otimes N}$ and hence the initial error can be computed as:

$$\frac{1}{N}\mathscr{F}^{(N)}(\mu_0^{(N)}) - \mathscr{F}(\pi)$$

$$= \mathbb{E}[F_0(\mu_{\mathscr{X}_0})] + \frac{\zeta}{2N}\mathbb{E}_{\mathscr{X}\sim\mu_0^{(N)}}\left[\sum_{i=1}^N \|\boldsymbol{x}^{(i)}\|^2\right] - \text{Ent}(\mu_0) - F_0(\pi) - \frac{\zeta}{2}\mathbb{E}_\pi[\|\boldsymbol{x}\|^2] + \text{Ent}(\pi)$$

$$= \mathbb{E}[F_0(\mu_{\mathscr{X}_0})] - F_0(\pi) + \frac{\zeta}{2}\mathbb{E}_{\boldsymbol{x}\sim\mu_0}\left[\|\boldsymbol{x}\|^2\right] - \text{Ent}(\mu_0) - \text{KL}\left(\pi\|\gamma_\zeta\right)$$

$$\leq \mathbb{E}[F_0(\mu_{\mathscr{X}_0})] - F_0(\pi) + \frac{\zeta}{2}\mathbb{E}_{\boldsymbol{x}\sim\mu_0}\left[\|\boldsymbol{x}\|^2\right] - \text{Ent}(\mu_0) := C_{\mu_0}. \tag{B.8}$$

Here, $\gamma_\zeta := \mathcal{N}(0, \frac{1}{\zeta}I_d)$. The proof of [Theorem 3.3]{.underline} can thus be concluded by the following relation proved in Lemma 1 of [68]:

$$\frac{\sigma}{N}\text{KL}(\mu_T^{(N)}\|\pi^{\otimes N}) \leq \frac{1}{N}\mathbb{E}\left[\mathscr{F}^{(N)}(\mu_T^{(N)})\right] - \mathscr{F}(\pi).$$

### B.3. Proof of [Proposition B.1]{.underline}

*Proof.* Recall below the forms of $\nabla F_0'[\mu_{\mathscr{X}}]$ in Eq. (6) and $\nabla F_0'[\mu_{\bar{\mathscr{X}}}]$ in Eq. (7):

$$\nabla F_0'[\mu_{\mathscr{X}}](\boldsymbol{x}) = \mathbb{E}_{u\sim\rho}\left[R_1'\left(\frac{1}{N}\sum_{j=1}^N q_1(u, \boldsymbol{x}^{(j)})\right)\nabla_2 q_1(u, \boldsymbol{x})\right] + \frac{1}{N}\sum_{j=1}^N \nabla_1 q_2(\boldsymbol{x}, \boldsymbol{x}^{(j)}),$$

$$\nabla F_0'[\mu_{\bar{\mathscr{X}}}](\boldsymbol{x}) = \mathbb{E}_{u\sim\rho}\left[R_1'\left(\frac{1}{M}\sum_{j=1}^M q_1(u, \bar{\boldsymbol{x}}^{(j)})\right)\nabla_2 q_1(u, \boldsymbol{x})\right] + \frac{1}{M}\sum_{j=1}^M \nabla_1 q_2(\boldsymbol{x}, \bar{\boldsymbol{x}}^{(j)}).$$

Consider the difference of the above. For any $m \in \{1, \ldots, d\}$, noting that $\nabla_2 q_1(u, \boldsymbol{x})$ is uniformly bounded, we obtain

$$\left|\left[\nabla F_0'[\mu_{\mathscr{X}}](\boldsymbol{x})\right]_m - \left[\nabla F_0'[\mu_{\bar{\mathscr{X}}}](\boldsymbol{x})\right]_m\right| \leq Q_1\mathbb{E}_{u\sim\rho}\left[\left|R_1'\left(\frac{1}{N}\sum_{j=1}^N q_1(u, \boldsymbol{x}^{(j)})\right) - R_1'\left(\frac{1}{M}\sum_{j=1}^M q_1(u, \bar{\boldsymbol{x}}^{(j)})\right)\right|\right]$$

$$+ \left|\frac{1}{N}\sum_{j=1}^N \partial_{1,m}q_2(\boldsymbol{x}, \boldsymbol{x}^{(j)}) - \frac{1}{M}\sum_{j=1}^M \partial_{1,m}q_2(\boldsymbol{x}, \bar{\boldsymbol{x}}^{(j)})\right|$$

$$\leq Q_1 L_R\mathbb{E}_{u\sim\rho}\left|\frac{1}{N}\sum_{j=1}^N q_1(u, \boldsymbol{x}^{(j)}) - \frac{1}{M}\sum_{j=1}^M q_1(u, \bar{\boldsymbol{x}}^{(j)})\right| + \left|\frac{1}{N}\sum_{j=1}^N \partial_{1,m}q_2(\boldsymbol{x}, \boldsymbol{x}^{(j)}) - \frac{1}{M}\sum_{j=1}^M \partial_{1,m}q_2(\boldsymbol{x}, \bar{\boldsymbol{x}}^{(j)})\right|.$$

Here, $\partial_{1,m}q_2$ denotes taking the $m$-th partial derivative with respect to the first argument of $q_2$.

By [76, Example 3], there exists an event $\mathcal{E}(\delta)$ of probability at most $\delta/2$ on which the KT-SPLIT-Compress($\delta$) coreset is $\nu_0 \|f\|_{\mathcal{H}}$-sub-Gaussian for each $f$ in the RKHS. Here,

$$\nu_0 \leq \frac{2}{\sqrt{N}\sqrt{3}}\sqrt{\log\left(\frac{6\sqrt{N}\log_4 N}{\delta}\right)\kappa}.$$

In other words, for every $f$, there exists a $\nu_0\|f\|_{\mathcal{H}}$ sub-Gaussian random variable such that $X_f$ equals the KT-SPLIT-Compress($\delta$) coreset $f$-integration error on $\mathcal{E}(\delta)$. Furthermore, for any $\nu_0\|f\|_{\mathcal{H}}^2$-sub-Gaussian variable $X$, $\mathbb{E}[X^2] \leq \nu_0^2\|f\|_{\mathcal{H}}^2$ [11, Lem. 1.2], so $\mathbb{E}[X_f^2\mathbb{1}_{\mathcal{E}(\delta)}] \leq \mathbb{E}[X_f^2] \leq \nu_0^2\|f\|_{\mathcal{H}}^2$. Finally we apply this bound to the appropriate test function in our $\mathcal{H}$ for each component $j$ and $x$ and sum the results.

From Assumption 3.2, we know that both $q_1(u,\cdot) \in \mathcal{H}_k$ and $\partial_{1,m}q_2(\boldsymbol{x},\cdot) \in \mathcal{H}_k$ for any $m \in \{1,\ldots,d\}$. Hence,

$$\mathbb{E}\left[\left|\frac{1}{N}\sum_{j=1}^{N}q_1(u,\boldsymbol{x}^{(j)}) - \frac{1}{M}\sum_{j=1}^{M}q_1(u,\bar{\boldsymbol{x}}^{(j)})\right|^2 \cdot \mathbb{1}_{\mathcal{E}(\delta)}\right] \leq \nu_0^2\mathscr{Q}_1^2 \quad \text{and}$$

$$\mathbb{E}\left[\left|\frac{1}{N}\sum_{j=1}^{N}\partial_{1,m}q_2(\boldsymbol{x},\boldsymbol{x}^{(j)}) - \frac{1}{M}\sum_{j=1}^{M}\partial_{1,m}q_2(\boldsymbol{x},\bar{\boldsymbol{x}}^{(j)})\right|^2 \cdot \mathbb{1}_{\mathcal{E}(\delta)}\right] \leq \nu_0^2\mathscr{Q}_2^2.$$

Therefore, we combined the above two inequalities, for any $\boldsymbol{x} \in \mathbb{R}^d$,

$$\mathbb{E}\left[\left|\left[\nabla F_0'[\mu_{\mathscr{X}}](\boldsymbol{x})\right]_m - \left[\nabla F_0'[\mu_{\bar{\mathscr{X}}}](\boldsymbol{x})\right]_m\right|^2 \cdot \mathbb{1}_{\mathcal{E}(\delta)}\right] \leq (Q_1^2 L_R^2 + 1)\nu_0^2(\mathscr{Q}_1^2 + \mathscr{Q}_2^2).$$

Summing the above from $m = 1$ to $d$, we obtain, for any $\boldsymbol{x} \in \mathbb{R}^d$,

$$\mathbb{E}\left[\|\nabla F_0'[\mu_{\mathscr{X}}](\boldsymbol{x}) - \nabla F_0'[\mu_{\bar{\mathscr{X}}}](\boldsymbol{x})\|^2 \cdot \mathbb{1}_{\mathcal{E}(\delta)}\right] \leq d(Q_1^2 L_R^2 + 1)\nu_0^2(\mathscr{Q}_1^2 + \mathscr{Q}_2^2).$$

Define $C := \frac{4}{3}(Q_1^2 L_R^2 + 1)(\mathscr{Q}_1^2 + \mathscr{Q}_2^2)\kappa$ and the proof is concluded. $\qquad\square$

## B.4. Proof of Proposition B.2

*Proof.* The proof is an adaptation of the proof of Theorem 2 in [68]. Assumption 1 in [68] holds with $C_1 = |R_1'(0)| + L_R Q_1 + Q_2$ and $C_2 = Q_1^2 L_R + Q_2 > 0$ proved in Lemma B.3; Assumption 2 in [68] holds with $\bar{\alpha} = \frac{\zeta}{2\sigma}\exp(-\frac{4C_3}{\zeta\sigma}\sqrt{\frac{2d}{\pi}})$ following the same Miclo's trick [6, Lemma 2.1]; Assumption 3 in [68] holds since $R_1$ is convex, as proved in Lemma B.4; Assumption 4 in [68] holds with $B = 2L_R Q_1^2 + Q_2$ proved in Lemma B.6.

Recall the update scheme of the thinned MFLD at iteration $t$: for $i = 1,\ldots,N$,

$$\boldsymbol{x}_{t+1}^{(i)} = \boldsymbol{x}_t^{(i)} - \gamma\left\{v(\boldsymbol{x}_t^{(i)};\{\boldsymbol{x}_t^{(j)}\}_{j=1}^N) + \omega(\boldsymbol{x}_t^{(i)};\{\boldsymbol{x}_t^{(j)}\}_{j=1}^N,\{\bar{\boldsymbol{x}}_t^{(j)}\}_{j=1}^M) + \zeta\boldsymbol{x}_t^{(i)}\right\} + \sqrt{2\sigma\gamma}\,\xi_t^{(i)}, \tag{B.9}$$

where $v : \mathbb{R}^d \to \mathbb{R}^d$ denotes the velocity field of the original mean field Langevin dynamics defined in Eq. (B.4) and $\omega : \mathbb{R}^d \to \mathbb{R}^d$ denotes the additional approximation error induced by using a thinned coreset defined in Eq. (B.5). To aid analysis, we construct another $N$-particle update scheme which is a continuous one-step interpolation for $t$-th iteration of the original MFLD: for $i = 1,\ldots,N$,

$$\mathrm{d}\boldsymbol{y}_r^{(i)} = -\left\{v\left(\boldsymbol{y}_0^{(i)};\{\boldsymbol{y}_0^{(i)}\}_{i=1}^N\right) + \omega\left(\boldsymbol{y}_0^{(i)};\{\boldsymbol{y}_0^{(i)}\}_{i=1}^N,\{\bar{\boldsymbol{y}}_0^{(i)}\}_{i=1}^M\right) + \zeta\boldsymbol{y}_0^{(i)}\right\}\mathrm{d}r + \sqrt{2\sigma}\,\mathrm{d}B_r,$$

$$\text{where}\quad \{\boldsymbol{y}_0^{(i)}\}_{i=1}^N = \{\boldsymbol{x}_t^{(i)}\}_{i=1}^N, \quad \{\bar{\boldsymbol{y}}_0^{(i)}\}_{i=1}^M = \{\bar{\boldsymbol{x}}_t^{(i)}\}_{i=1}^M,$$

and $B_r$ is the standard Brownian motion on $\mathbb{R}^d$. Since the drift term in the above stochastic difference equation is time-independent, it admits the following solution: for $i = 1,\ldots,N$ and any $r \in [0,\gamma]$,

$$\boldsymbol{y}_r^{(i)} = \boldsymbol{y}_0^{(i)} - r\left\{v\left(\boldsymbol{y}_0^{(i)};\{\boldsymbol{y}_0^{(i)}\}_{i=1}^N\right) + \omega\left(\boldsymbol{y}_0^{(i)};\{\boldsymbol{y}_0^{(i)}\}_{i=1}^N,\{\bar{\boldsymbol{y}}_0^{(i)}\}_{i=1}^M\right) + \zeta\boldsymbol{y}_0^{(i)}\right\} + \sqrt{2\sigma r}\,\xi_r^{(i)}. \tag{B.10}$$

Therefore, comparing Eq. (B.10) and Eq. (B.9), we see that $\{\boldsymbol{y}_\gamma^{(i)}\}_{i=1}^N$ coincide with $\{\boldsymbol{x}_{t+1}^{(i)}\}_{i=1}^N$. Denote by $\nu_r$ the joint distribution of the above $N$ particles $\{\boldsymbol{y}_r^{(i)}\}_{i=1}^N$ at time $r \in [0, \gamma]$. Recall that $\pi^{\otimes N}$ denotes the $N$-fold product measure of the target distribution $\pi$. Following the same derivations as (27)-(30) in [65], we obtain

$$\frac{\mathrm{d}\mathscr{F}^{(N)}}{\mathrm{d}r}(\nu_r) \leq -\frac{\sigma^2}{2} \int \nu_r(\boldsymbol{y}) \left\| \nabla \log \frac{\nu_r}{\pi^{\otimes N}}(\boldsymbol{y}) \right\|^2 \mathrm{d}\boldsymbol{y} + \mathbb{E}_{\nu_0, \nu_r}\left[ \sum_{i=1}^N \left\| v(\boldsymbol{y}_0^{(i)}) - v(\boldsymbol{y}_r^{(i)}) \right\|_2^2 \right]$$

$$+ \sum_{i=1}^N \mathbb{E}\left[ \left\| \omega\left( \boldsymbol{y}_0^{(i)}; \{\boldsymbol{y}_0^{(j)}\}_{j=1}^N, \{\bar{\boldsymbol{y}}_0^{(j)}\}_{j=1}^M \right) \right\|^2 \right]$$

$$\leq -\frac{\sigma^2}{2} \int \nu_r(\boldsymbol{y}) \left\| \nabla \log \frac{\nu_r}{\pi^{\otimes N}}(\boldsymbol{y}) \right\|^2 \mathrm{d}\boldsymbol{y} + N\delta_\gamma + Nc_0 \frac{(\log N)^3}{N}.$$

The last inequality holds by applying the derivations on the page of 15 in [65] to the second term and by applying Eq. (B.7) on the third term. Here, $\delta_\gamma = \mathcal{O}(\gamma^2 + \gamma \sigma d)$ represents the time discretization error. Next, noticing that the first term above is the Fisher divergence between $\nu_r$ and $\pi^{\otimes N}$, we apply Lemma 2 in [68] which gives

$$\frac{\mathrm{d}\mathscr{F}^{(N)}}{\mathrm{d}r}(\nu_r) \leq -\bar{\alpha}\sigma \left( \mathscr{F}^{(N)}(\nu_r) - N\mathscr{F}(\pi) - B \right) + N\delta_\gamma + c_0(\log N)^3.$$

Since $\nu_\gamma = \mu_{t+1}$ and $\nu_0 = \mu_t$, we apply the Gronwall's lemma [36, Appendix B.2, Gronwall's inequality] and obtain

$$\mathscr{F}^{(N)}(\mu_{t+1}^{(N)}) - N\mathscr{F}(\pi) - B - \frac{N\delta_\gamma + c_0(\log N)^3}{2\bar{\alpha}}$$

$$\leq \exp(-\bar{\alpha}\sigma\gamma)\left( \mathscr{F}^{(N)}(\mu_t^{(N)}) - N\mathscr{F}(\pi) - B - \frac{N\delta_\gamma + c_0(\log N)^3}{2\bar{\alpha}} \right).$$

The above inequality holds at every iteration, hence, we arrive at the desired result and conclude the proof. $\qquad\square$

### B.5. Auxiliary Lemmas

**Lemma B.3** (Bounded and Lipschitz vector field). *Suppose Assumption 3.1 holds. There exists constants $C_1 = (|R_1'(0)| + L_R Q_1)Q_1 + Q_2$ and $C_2 = (|R_1'(0)| + L_R Q_1 + Q_2)Q_1 + Q_1^2 L_R + Q_2 > 0$, such that the following properties hold: 1) for any $\mu \in \mathcal{P}_2(\mathbb{R}^d)$ and $\boldsymbol{x} \in \mathbb{R}^d$, we have $\|\nabla F_0'[\mu](\boldsymbol{x})\| \leq C_1$. 2) for any $\mu, \mu^\circ \in \mathcal{P}_2(\mathbb{R}^d), \boldsymbol{x}, \boldsymbol{x}^\circ \in \mathbb{R}^d$, $\|\nabla F_0'[\mu](\boldsymbol{x}) - \nabla F_0'[\mu^\circ](\boldsymbol{x}^\circ)\| \leq C_2(W_2(\mu, \mu^\circ) + \|\boldsymbol{x} - \boldsymbol{x}^\circ\|)$, where $W_2$ is the Wasserstein-2 distance.*

*Proof.* From Assumption 3.1, we have $R_1'\left(\int q_1(u, \boldsymbol{x})\mathrm{d}\mu(\boldsymbol{x})\right) \leq |R_1'(0)| + L_R |\int q_1(u, \boldsymbol{x})\mathrm{d}\mu(\boldsymbol{x})| \leq |R_1'(0)| + L_R Q_1$. So we have, for any $\mu \in \mathcal{P}_2(\mathbb{R}^d)$ and any $\boldsymbol{x} \in \mathbb{R}^d$,

$$\nabla F_0'[\mu](\boldsymbol{x}) = \mathbb{E}_{u\sim\rho}\left[ R_1'\left( \int q_1(u, \boldsymbol{x})\mathrm{d}\mu(\boldsymbol{x}) \right) \nabla_2 q_1(u, \boldsymbol{x}) \right] + \int \nabla_1 q_2(\boldsymbol{x}, \tilde{\boldsymbol{x}})\mathrm{d}\mu(\tilde{\boldsymbol{x}})$$

$$\leq (|R_1'(0)| + L_R Q_1) Q_1 + Q_2.$$

So we have proved property (i). Also, we have for any $\mu, \mu^\circ \in \mathcal{P}_2(\mathbb{R}^d)$ and any $\boldsymbol{x}, \boldsymbol{x}^\circ \in \mathbb{R}^d$,

$$\|\nabla F_0'[\mu](\boldsymbol{x}) - \nabla F_0'[\mu^\circ](\boldsymbol{x}^\circ)\|$$

$$\leq \left\| \mathbb{E}_{u\sim\rho}\left[ R_1'\left( \int q_1(u, \boldsymbol{x})\mathrm{d}\mu(\boldsymbol{x}) \right) \nabla_2 q_1(u, \boldsymbol{x}) \right] - \mathbb{E}_{u\sim\rho}\left[ R_1'\left( \int q_1(u, \boldsymbol{x})\mathrm{d}\mu^\circ(\boldsymbol{x}) \right) \nabla_2 q_1(u, \boldsymbol{x}^\circ) \right] \right\|$$

$$+ \left\| \int \nabla_1 q_2(\boldsymbol{x}, \tilde{\boldsymbol{x}})\mathrm{d}\mu(\tilde{\boldsymbol{x}}) - \int \nabla_1 q_2(\boldsymbol{x}^\circ, \tilde{\boldsymbol{x}})\mathrm{d}\mu^\circ(\tilde{\boldsymbol{x}}) \right\|$$

$$\leq (|R_1'(0)| + L_R Q_1)\mathbb{E}_{u\sim\rho}\left[ \|\nabla_2 q_1(u, \boldsymbol{x}) - \nabla_2 q_1(u, \boldsymbol{x}^\circ)\| \right] + Q_1 L_R \mathbb{E}_{u\sim\rho}\left[ \left\| \int q_1(u, \boldsymbol{x})\mathrm{d}(\mu^\circ - \mu)(\boldsymbol{x}) \right\| \right]$$

$$+ \left\| \int \nabla_1 q_2(\boldsymbol{x}, \tilde{\boldsymbol{x}})\mathrm{d}\mu(\tilde{\boldsymbol{x}}) - \int \nabla_1 q_2(\boldsymbol{x}^\circ, \tilde{\boldsymbol{x}})\mathrm{d}\mu(\tilde{\boldsymbol{x}}) \right\| + \left\| \int \nabla_1 q_2(\boldsymbol{x}^\circ, \tilde{\boldsymbol{x}})\mathrm{d}(\mu - \mu^\circ)(\tilde{\boldsymbol{x}}) \right\|$$

$$\leq (|R_1'(0)| + L_R Q_1)Q_1\|\boldsymbol{x} - \boldsymbol{x}^\circ\| + Q_1^2 L_R W_2(\mu, \mu^\circ) + Q_2\|\boldsymbol{x} - \boldsymbol{x}^\circ\| + Q_2 W_2(\mu, \mu^\circ)$$

$$= (|R_1'(0)| + L_R Q_1 + Q_2) Q_1 \|\boldsymbol{x} - \boldsymbol{x}^\circ\| + (Q_1^2 L_R + Q_2) W_2(\mu, \mu^\circ).$$

The proof is thus concluded. The second last inequality holds because, if $g : \mathbb{R}^d \to \mathbb{R}$ is a $L$-Lipschitz function, then $\left\| \int g(\boldsymbol{x}) \mathrm{d}(\mu - \mu^\circ)(\boldsymbol{x}) \right\| \le L W_1(\mu, \mu^\circ) \le L W_2(\mu, \mu^\circ).$ □

**Lemma B.4** (Convexity). *Suppose Assumption 3.1 holds. The functional $F_0 : \mathcal{P}_2(\mathbb{R}^d) \to \mathbb{R}$ is convex, i.e., for all $\vartheta \in (0, 1)$ and for all $\mu, \nu \in \mathcal{P}_2(\mathbb{R}^d)$, there is $F_0(\vartheta \mu + (1 - \vartheta)\nu) \le \vartheta F_0(\mu) + (1 - \vartheta) F_0(\nu).$*

*Proof.* The proof is concluded by noticing that the first component of $F$ is a composition of a convex function $R_1$ and a linear functional $\mu \mapsto \int q_1(u, \boldsymbol{x}) \mathrm{d}\mu(\boldsymbol{x})$; and noticing that the second component of $F$ is a quadratic functional of $\mu$ and hence convex. □

**Definition B.5** (Bregman divergence). *For any $\mu, \mu^\circ \in \mathcal{P}_2(\mathbb{R}^d)$, define the Bregman divergence of the functional $F$ as $B_F(\mu, \mu^\circ) := F(\mu) - F(\mu^\circ) - \int F'[\mu^\circ](\boldsymbol{x}) \mathrm{d}(\mu - \mu^\circ)(\boldsymbol{x}).$*

**Lemma B.6** (Leave-one-out stability). *Suppose Assumption 3.1 holds. There exists a constant $B = 2 L_R Q_1^2 + Q_2$ such that for any $\mathscr{X} = \{\boldsymbol{x}^{(i)}\}_{i=1}^N \in (\mathbb{R}^d)^N$, any $\boldsymbol{x} \in \mathbb{R}^d$, and any $i \in \{1, 2, \ldots, N\}$,*

$$B_{F_0}\left( \frac{1}{N} \sum_{i=1}^N \delta_{\boldsymbol{x}^{(i)}}, \frac{1}{N} \sum_{i=1}^N \delta_{\boldsymbol{x}^{(i)}} - \frac{1}{N} \delta_{\boldsymbol{x}^{(i)}} + \frac{1}{N} \delta_{\boldsymbol{x}} \right) \le \frac{B}{N^2}.$$

*Proof.* Recall functional $F_0$ of the form $F_0(\mu) = \mathbb{E}_{u \sim \rho}[R_1\left(\int q_1(u, \boldsymbol{x}) \mathrm{d}\mu(\boldsymbol{x})\right)] + \iint q_2(\boldsymbol{x}, \tilde{\boldsymbol{x}}) \mathrm{d}\mu(\boldsymbol{x}) \mathrm{d}\mu(\tilde{\boldsymbol{x}})$ in Eq. (3). We consider the Bregman divergence of the two components separately. Denote the first component by $F_1(\mu) = \mathbb{E}_{u \sim \rho}[R_1\left(\int q_1(u, \boldsymbol{x}) \mathrm{d}\mu(\boldsymbol{x})\right)]$.

$$B_{F_1}\left( \frac{1}{N} \sum_{j=1}^N \delta_{\boldsymbol{x}^{(j)}}, \frac{1}{N} \sum_{j \neq i}^N \delta_{\boldsymbol{x}^{(j)}} + \frac{1}{N} \delta_{\boldsymbol{x}} \right)$$

$$= \mathbb{E}_{u \sim \rho}\left[ R_1\left( \frac{1}{N} \sum_{j=1}^N q_1(u, \boldsymbol{x}^{(j)}) \right) \right] - \mathbb{E}_{u \sim \rho}\left[ R_1\left( \frac{1}{N} \sum_{j \neq i}^N q_1(u, \boldsymbol{x}^{(j)}) + \frac{1}{N} q_1(u, \boldsymbol{x}) \right) \right]$$

$$- \mathbb{E}_{u \sim \rho}\left[ R_1'\left( \frac{1}{N} \sum_{j \neq i}^N q_1(u, \boldsymbol{x}^{(j)}) + \frac{1}{N} q_1(u, \boldsymbol{x}) \right) \left( \frac{1}{N} q_1(u, \boldsymbol{x}^{(i)}) - \frac{1}{N} q_1(u, \boldsymbol{x}) \right) \right]$$

$$\le \frac{L_R}{2} \mathbb{E}_{u \sim \rho}\left[ \left( \frac{1}{N} q_1(u, \boldsymbol{x}^{(i)}) - \frac{1}{N} q_1(u, \boldsymbol{x}) \right)^2 \right] \le \frac{2 L_R Q_1^2}{N^2}.$$

The second to last inequality holds because $R_1$ is $L_R$-Lipschitz. Hence $R_1(y) - R_1(y^\circ) - R_1'(y^\circ)(y - y^\circ) \le \frac{L_R}{2}(y - y^\circ)^2$ for any $y, y^\circ \in \mathbb{R}$. The last inequality holds by the uniform bound on $q_1$ in Assumption 3.1.

Next, we consider the second component in $F$, denoted by $F_2(\mu) = \iint q_2(\boldsymbol{x}, \tilde{\boldsymbol{x}}) \mathrm{d}\mu(\boldsymbol{x}) \mathrm{d}\mu(\tilde{\boldsymbol{x}})$. Note that the Bregman divergence equals

$$\iint q_2(\boldsymbol{x}, \tilde{\boldsymbol{x}}) \mathrm{d}\mu(\boldsymbol{x}) \mathrm{d}\mu(\tilde{\boldsymbol{x}}) - \iint q_2(\boldsymbol{x}, \tilde{\boldsymbol{x}}) \mathrm{d}\mu^\circ(\boldsymbol{x}) \mathrm{d}\mu^\circ(\tilde{\boldsymbol{x}}) - 2 \iint q_2(\boldsymbol{x}, \tilde{\boldsymbol{x}}) \mathrm{d}\mu^\circ(\tilde{\boldsymbol{x}}) \mathrm{d}(\mu - \mu^\circ)(\boldsymbol{x})$$

$$= \iint q_2(\boldsymbol{x}, \tilde{\boldsymbol{x}}) \mathrm{d}\mu(\boldsymbol{x}) \mathrm{d}\mu(\tilde{\boldsymbol{x}}) + \iint q_2(\boldsymbol{x}, \tilde{\boldsymbol{x}}) \mathrm{d}\mu(\boldsymbol{x}) \mathrm{d}\mu^\circ(\tilde{\boldsymbol{x}}) - 2 \iint q_2(\boldsymbol{x}, \tilde{\boldsymbol{x}}) \mathrm{d}\mu^\circ(\tilde{\boldsymbol{x}}) \mathrm{d}\mu(\boldsymbol{x})$$

$$= \iint q_2(\boldsymbol{x}, \tilde{\boldsymbol{x}}) \mathrm{d}(\mu - \mu^\circ)(\boldsymbol{x}) \mathrm{d}(\mu - \mu^\circ)(\tilde{\boldsymbol{x}}).$$

Therefore, from the uniform bound on $q_2$ in Assumption 3.1, we have

$$B_{F_2}\left( \frac{1}{N} \sum_{j=1}^N \delta_{\boldsymbol{x}^{(j)}}, \frac{1}{N} \sum_{j \neq i}^N \delta_{\boldsymbol{x}^{(j)}} + \frac{1}{N} \delta_{\boldsymbol{x}} \right) \le \frac{Q_2}{N^2}.$$

The proof is concluded by combining the above two bounds. □

**Lemma B.7** (Variance for random subsampling). *Let $X_1, \ldots, X_N$ be $N$ not necessarily i.i.d. random variables in $\mathbb{R}^d$, and let $\tilde{X}_1, \ldots, \tilde{X}_M$ be sampled i.i.d with replacement from $\{X_1, \ldots, X_N\}$. For any function $f : \mathbb{R}^d \to \mathbb{R}$ with $\mathrm{Var}(f(X_1)) < \infty$, we have*

$$\mathrm{Var}\left(\frac{1}{M}\sum_{j=1}^{M} f(\tilde{X}_j) - \frac{1}{N}\sum_{i=1}^{N} f(X_i)\right) = \frac{1}{M} \cdot \mathbb{E}\left[\frac{1}{N}\sum_{i=1}^{N}\left(f(X_i) - \frac{1}{N}\sum_{\ell=1}^{N} f(X_\ell)\right)^2\right].$$

*Proof.* Let

$$\Delta := \frac{1}{M}\sum_{j=1}^{M} f(\tilde{X}_j) - \frac{1}{N}\sum_{i=1}^{N} f(X_i).$$

Sampling with replacement means there exist i.i.d. indices $I_1, \ldots, I_M \sim \mathrm{Unif}\{1, \ldots, N\}$, independent of $(X_i)_{i=1}^N$, such that $\tilde{X}_j = X_{I_j}$. Conditioning on $(X_i)_{i=1}^N$, the variables $f(X_{I_1}), \ldots, f(X_{I_M})$ are i.i.d. with mean $\frac{1}{N}\sum_{i=1}^N f(X_i)$, hence $\mathbb{E}[\Delta \mid X_{1:N}] = 0$ and

$$\mathrm{Var}(\Delta \mid X_{1:N}) = \mathrm{Var}\left(\frac{1}{M}\sum_{j=1}^{M} f(X_{I_j}) \,\Big|\, X_{1:N}\right) = \frac{1}{M}\left(\frac{1}{N}\sum_{i=1}^{N} f(X_i)^2 - \left(\frac{1}{N}\sum_{i=1}^{N} f(X_i)\right)^2\right)$$

$$= \frac{1}{M} \cdot \frac{1}{N}\sum_{i=1}^{N}\left(f(X_i) - \frac{1}{N}\sum_{\ell=1}^{N} f(X_\ell)\right)^2.$$

By the law of total variance and $\mathbb{E}[\Delta \mid X_{1:N}] = 0$,

$$\mathrm{Var}(\Delta) = \mathbb{E}[\mathrm{Var}(\Delta \mid X_{1:N})] = \frac{1}{M}\mathbb{E}\left[\frac{1}{N}\sum_{i=1}^{N}\left(f(X_i) - \frac{1}{N}\sum_{\ell=1}^{N} f(X_\ell)\right)^2\right].$$

$\square$

**Remark B.8** (Variance for random subsampling and kernel thinning). *Lemma B.7 shows that, when $M = \lceil\sqrt{N}\rceil$, if the expected empirical variance of $\{f(X_i)\}_{i=1}^N$ is uniformly bounded away from zero and infinity, the variance induced by random thinning scales as $\Omega(N^{-\frac{1}{2}})$. Substituting this rate into Theorem 2 of [79] yields an additional error term of order $\Omega(N^{-\frac{1}{2}})$ in the convergence upper bound of MFLD with random thinning at each iteration. In contrast, the additional error introduced by* KT-SPLIT-*Compress scales as $\tilde{\mathcal{O}}(N^{-1})$ as proved in Theorem 3.3.*

**Lemma B.9** (Pro posterior). *Let $\varkappa : \mathcal{Y} \times \mathcal{Y} \to \mathbb{R}$ be a bounded symmetric positive semi-definite reproducing kernel. Let $q_2(\boldsymbol{x}, \boldsymbol{x}^\circ) = \iint \varkappa(y, y')p_{\boldsymbol{x}}(y)p_{\boldsymbol{x}^\circ}(y')\mathrm{d}y\mathrm{d}y'$. Suppose that $\iint |\varkappa(y, y')| \sup_{\boldsymbol{x}\in\mathbb{R}^d} |p_{\boldsymbol{x}}(y)|\mathrm{d}y\mathrm{d}y' < \infty$ and that $\iint |\varkappa(y, y')| \sup_{\boldsymbol{x}\in\mathbb{R}^d} \|\nabla_{\boldsymbol{x}}p_{\boldsymbol{x}}(y)\|\mathrm{d}y\mathrm{d}y' < \infty$. Suppose that the mapping $\boldsymbol{x} \mapsto p_{\boldsymbol{x}}(y) \in \mathcal{H}_k$ with $\mathcal{H}_k$ being an RKHS associated with another bounded kernel $k : \mathbb{R}^d \times \mathbb{R}^d \to \mathbb{R}$ and $\sup_{y\in\mathcal{Y}} \|\boldsymbol{x} \mapsto p_{\boldsymbol{x}}(y)\|_{\mathcal{H}_k} < \infty$. Then, we have $\nabla_1 q_2(\boldsymbol{x}, \cdot) \in \mathcal{H}_k^{\otimes d}$ for all $\boldsymbol{x}$.*

*Proof.* First, we want to justify that we can interchange derivative and integration here:

$$\nabla_{\boldsymbol{x}} \iint \varkappa(y, y')p_{\boldsymbol{x}}(y)\mathrm{d}y\mathrm{d}y' = \iint \varkappa(y, y')\nabla_{\boldsymbol{x}}p_{\boldsymbol{x}}(y)\mathrm{d}y\mathrm{d}y'.$$

To this end, we are going to apply Theorem A.7 in [32]. Since $\iint \sup_{\boldsymbol{x}\in\mathbb{R}^d} |\varkappa(y, y')p_{\boldsymbol{x}}(y)|\mathrm{d}y\mathrm{d}y' < \infty$, so (A.2a) in [32] is satisfied; since $\iint |\varkappa(y, y')| \sup_{\boldsymbol{x}\in\mathbb{R}^d} \|\nabla_{\boldsymbol{x}}p_{\boldsymbol{x}}(y)\|\mathrm{d}y\mathrm{d}y' < \infty$, so (A.2b) and (A.8) in [32] is satisfied. Therefore, all conditions in Theorem A.7 in [32] hold, and we have

$$\nabla_1 q_2(\boldsymbol{x}, \boldsymbol{x}^\circ) = \iint \varkappa(y, y')\nabla_{\boldsymbol{x}}p_{\boldsymbol{x}}(y)p_{\boldsymbol{x}^\circ}(y') \,\mathrm{d}y\mathrm{d}y', \quad \forall \ell \in \{1, \ldots, d\}.$$

From the Bochner intergrability of $(y, y') \mapsto \varkappa(y, y')\nabla_{\boldsymbol{x}}p_{\boldsymbol{x}}(y)p_\cdot(y')$ for any $\boldsymbol{x} \in \mathbb{R}^d$ with respect to the Lebesgue measure over $\mathcal{Y} \times \mathcal{Y}$ [78, Definition A.5.20], we have

$$\|\nabla_1 q_2(\boldsymbol{x}, \cdot)\|_{\mathcal{H}_k^{\otimes d}} = \left\|\iint \varkappa(y, y')\nabla_{\boldsymbol{x}}p_{\boldsymbol{x}}(y)p_\cdot(y')\mathrm{d}y\mathrm{d}y'\right\|_{\mathcal{H}_k^{\otimes d}} \leq \iint |\varkappa(y, y')|\|\nabla_{\boldsymbol{x}}p_{\boldsymbol{x}}(y)\| \|p_\cdot(y')\|_{\mathcal{H}_k} \mathrm{d}y\mathrm{d}y'$$

$$\leq \sup_{y'} \|p_{\cdot}(y')\|_{\mathcal{H}_k} \cdot \iint |\varkappa(y,y')| \|\nabla_{\boldsymbol{x}} p_{\boldsymbol{x}}(y)\| \mathrm{d}y \mathrm{d}y' < \infty.$$

So we have concluded the proof that $\nabla_1 q_2(\boldsymbol{x}, \cdot) \in \mathcal{H}^{\otimes d}$ for all $\boldsymbol{x}$. $\qquad \square$

**Lemma B.10** (RKHS norm of mean-field neural network). *Let* $\{z_s\}_{s=1}^n$ *be fixed. Define* $k(\boldsymbol{x}, \boldsymbol{x}^{\circ}) = \sum_{s=1}^n \Psi(z_s, \boldsymbol{x}) \Psi(z_s, \boldsymbol{x}^{\circ})$, *and let* $\mathcal{H}_k$ *be its associated finite-rank RKHS. Then,* $\|\Psi(z_s, \cdot)\|_{\mathcal{H}_k} \leq 1$ *for any* $s \in \{1, \dots, n\}$.

*Proof.* Since $\{z_s\}_{s=1}^n$ is fixed, the kernel $k$ is deterministic. By the finite-rank construction of $k$, every $f \in \mathcal{H}_k$ can be written as $f(\cdot) = \sum_{r=1}^n a_r \Psi(z_r, \cdot)$, with RKHS norm $\|f\|_{\mathcal{H}_k} = \inf \{\|a\|_2 : f(\cdot) = \sum_{r=1}^n a_r \Psi(z_r, \cdot)\}$. In particular, taking $f(\cdot) = \Psi(z_s, \cdot)$, the coefficient vector $a = e_s$ is an admissible representation, where $e_s$ is the $s$-th standard basis vector in $\mathbb{R}^n$. Hence

$$\|\Psi(z_s, \cdot)\|_{\mathcal{H}_k} = \inf \left\{ \|a\|_2 : \Psi(z_s, \cdot) = \sum_{r=1}^n a_r \Psi(z_r, \cdot) \right\} \leq \|e_s\| = 1.$$

Therefore $\|\Psi(z_s, \cdot)\|_{\mathcal{H}_k} \leq 1$. $\qquad \square$

## C. Additional Experimental Details

Here, we provide additional experimental details for Section 4.

**Mean Field Neural Networks** The teacher network is defined as $h_{\text{teacher}}(z) = \frac{1}{N_0} \sum_{j=1}^{N_0} \Psi(z, \boldsymbol{w}^{(j)})$, where we set the number of teacher neurons to $N_0 = 100$. The parameters $\{\boldsymbol{w}^{(j)}\}_{j=1}^{N_0}$ are sampled independently from a mixture of 10 Gaussian distributions in $\mathbb{R}^{12}$ in order to induce heterogeneity across neurons. Specifically, we first sample a fixed set of Gaussian mode centers from $\mathcal{N}(0, 4)$. Each teacher neuron is then independently assigned to one of these modes, and its parameters are obtained by adding small isotropic Gaussian noise drawn from $\mathcal{N}(0, 0.2)$ around the corresponding mode center. Each parameter vector $\boldsymbol{w}^{(j)}$ consists of a first-layer weight $\boldsymbol{w}_1^{(j)} \in \mathbb{R}^{d-1}$, a first-layer bias $\boldsymbol{b}_1^{(j)} \in \mathbb{R}$, and a second-layer weight $\boldsymbol{w}_2^{(j)} \in \mathbb{R}$. The first- and second-layer weights are rescaled by a common factor of 0.8 to control the overall magnitude of the network output. In this experiment, we take $\Psi(z, \boldsymbol{w}) = \boldsymbol{w}_2 a(\boldsymbol{w}_1^\top z + \boldsymbol{b}_1)$ where the activation $a$ is ReLU. To ensure that Assumption 3.1 is satisfied, we apply a smooth clipping to the network output at levels $\pm 1e3$.

The dataset is generated with additive Gaussian noise corruption: $y = h_{\text{teacher}}(z) + \varsigma$ with $\varsigma \sim \mathcal{N}(0, 10^{-4})$, where the covariates $z$ are drawn from a Gaussian distribution $\mathcal{N}(0, 10^{-4})$. The student network shares the same functional form as the teacher, $h_{\text{student}}(z) = \frac{1}{N} \sum_{j=1}^N \Psi(z, \boldsymbol{x}^{(j)})$. The student parameters $\{\boldsymbol{x}^{(j)}\}_{j=1}^N$ are initialized independently from $\mathcal{N}(0, 0.05)$ and are subsequently trained using *noisy* gradient descent on the regularized empirical squared loss. During training, each update of a student particle uses a freshly generated batch of samples, corresponding to an online teacher–student learning setting [71, 5].

We employ $\ell_2$-regularization with strength $\zeta = 10^{-4}$, and inject additive Gaussian noise of scale $\sigma = 10^{-3}$ at each iteration. For a consistent comparison across methods, we use the same step size $\gamma = 0.01$ throughout all baselines. All methods are trained for 100 epochs.

The Sobolev kernel used in KT-MFLD is the sum of $k(\boldsymbol{x}, \boldsymbol{y}) = \sum_{s=1}^3 k(\boldsymbol{x}, \boldsymbol{y}; s)$ where for each $s \in \{1, 2, 3\}$,

$$k(\boldsymbol{x}, \boldsymbol{y}; s) := -1 + \prod_{j=1}^d \left( 1 + \frac{(-1)^{s-1}(2\pi)^{2s}}{(2s)!} B_{2s}(\{\boldsymbol{x}_j\} - \{\boldsymbol{y}_j\}) \right). \tag{C.1}$$

In the above definition, $B_{2s}$ denotes the Bernoulli polynomial of order $2s$ and $\{x\} = x - \lfloor x \rfloor$ denotes the fractional part of $x$. This kernel has been widely used in Quasi-Monte Carlo [29] and density estimation [48]. It admits an efficient Cython implementation in the `goodpoints` package. Specifically, KT-SPLIT-Compress corresponds to the function `goodpoints.compress.compress_kt` in the `goodpoints` package with the SKIP_SWAP flag set to TRUE, whereas KT-Compress corresponds to the same function with the SKIP_SWAP flag set to FALSE. The failure probability is set to be 0.5 for both KT-SPLIT-Compress and KT-Compress.

**Remark C.1** (On the $\Omega(N^2)$ cost of training mean-field neural networks). *In the context of mean-field neural networks, the update for each particle* $\boldsymbol{x}_t^{(i)}$ *at iterate* $t$ *is*

$$\mathbb{E}_{(z,y)\sim\rho}\left[ R_1'\left( \frac{1}{N} \sum_{j=1}^N q_1(z, \boldsymbol{x}_t^{(j)}), y \right) \nabla_2 q_1(z, \boldsymbol{x}_t^{(i)}) \right].$$

*When the data distribution is empirical $\rho = \frac{1}{n}\sum_{s=1}^{n}\delta_{(z_s, y_s)}$, a pre-computation strategy exists. Specifically, one may first compute $R_1'(\frac{1}{N}\sum_{j=1}^{N}q_1(z_s, \boldsymbol{x}_t^{(j)}), y_s)$ for all $s \in \{1, \ldots, n\}$ in $\Omega(N)$ time and save these quantities in memory. Given these precomputed values, the update for each particle can then be evaluated in $\mathcal{O}(1)$, yielding an overall $\mathcal{O}(N)$ cost per iteration. This strategy therefore reduces the cost from $\Omega(N^2)$ to $\mathcal{O}(N)$. However, this pre-computation strategy is not applicable in our online setting. In particular, for each particle at each iteration, a fresh batch of data samples is drawn from the teacher network, so the quantities above must be recomputed and cannot be shared across particles. Hence, the $\Omega(N^2)$ cost is explicit.*

**Quantization with MMD Minimization** We follow the experimental setup of [22, Section 5.1], where $\nu$ is chosen to be a mixture of Gaussian distributions on $\mathbb{R}^2$ and the goal is to find a good approximation of $\nu$ (in the sense of MMD) using a finite number of particles. The kernel $\varkappa$ used to compute the MMD is a Gaussian kernel with a fixed lengthscale 1, and the same kernel is used in both versions of KT-MFLD. The failure probability is set to be 0.5 for both KT-SPLIT-Compress and KT-Compress. It admits an efficient Cython implementation in the `goodpoints` package. For a consistent comparison across methods, we use the same step size $\gamma = 1.0$ throughout all baselines.

**PrO Posterior Inference** We consider the Lotka–Volterra model (LVM), which captures the cyclical dynamics arising from predation and reproduction [83]. The deterministic dynamics are governed by the coupled ordinary differential equations

$$\frac{\mathrm{d}u_1}{\mathrm{d}\tau} = \mathfrak{a}u_1 - \mathfrak{b}u_1u_2, \qquad u_1(0) = \xi_1,$$
$$\frac{\mathrm{d}u_2}{\mathrm{d}\tau} = \mathfrak{c}u_1u_2 - \mathfrak{d}u_2, \qquad u_2(0) = \xi_2,$$

where $u_1, u_2$ represent the prey and predator populations, respectively.

Observations are generated instead from a stochastic Lotka–Volterra model, in which the above dynamics are perturbed by stochastic noise, yielding a system of stochastic differential equations ($\epsilon_1 = \epsilon_2 = 0.4$):

$$\mathrm{d}u_1 = (\mathfrak{a}u_1 - \mathfrak{b}u_1u_2)\mathrm{d}\tau + \epsilon_1\mathrm{d}B_\tau, \qquad u_1(0) = \xi_1,$$
$$\mathrm{d}u_2 = (\mathfrak{c}u_1u_2 - \mathfrak{d}u_2)\mathrm{d}\tau + \epsilon_2\mathrm{d}B_\tau, \qquad u_2(0) = \xi_2,$$

Here, the additional stochasticity is used to represent the complexities of real predator-prey interactions that are not captured by the simple ODE model. The data-generating parameters that we used for the stochastic LVM were $\mathfrak{a} = \mathrm{logit}^{-1}(-2)$, $\mathfrak{b} = \mathrm{logit}^{-1}(-4)$, $\mathfrak{c} = 0.4$, $\mathfrak{d} = 0.02$. The initial prey and predator population were $\xi_1 = 10$ and $\xi_2 = 10$. The SDE model with intrinsic noise $\epsilon_1 = \epsilon_2 = 0.4$ was discretised for numerical simulation using the reversible Heun's method [49] with time step $\mathrm{d}\tau = 0.01$ and simulated from $\tau = 0$ to $\tau = 60$. Data were extracted at times $\tau_{1:n}$ ranging from 0 to 60 in time increments of 1.0, with Gaussian measurement noise of variance $\sigma^2 = 1$ added. The statistical model used $P_{\boldsymbol{x}}$ is outlined in the main text, where $\boldsymbol{u}_{\boldsymbol{x}}(\tau_i)$ is the outcome of the deterministic LVM simulated until time $\tau_i$. Specifically, the parameters $\boldsymbol{x} = [\boldsymbol{x}_1, \boldsymbol{x}_2]$ hence $d = 2$ are $\boldsymbol{x}_1 := \mathrm{logit}(\alpha)$ and $\boldsymbol{x}_2 = \mathrm{logit}(\beta/\alpha)$. The rest of the coefficients $\mathfrak{c}, \mathfrak{d}$ for the deterministic LVM remains the same as stochastic LVM.

The kernel used in both versions of KT-MFLD is the sum of $k(\boldsymbol{x}, \boldsymbol{y}) = \sum_{s=1}^{3} k(\boldsymbol{x}, \boldsymbol{y}; s)$ where $k(\boldsymbol{x}, \boldsymbol{y}; s)$ is the same as defined above in Eq. (C.1). The failure probability is set to be 0.5 for both KT-SPLIT-Compress and KT-Compress. The initial particles are drawn from $\mathcal{N}(0, 0.1)$.

**Mean-field Games** We solve the Hamilton–Jacobi–Bellman and continuity equations using a particle-based forward–backward sweep scheme under time discretizations with $\Delta t = 0.01$ within a fixed time interval of $[0, 1]$. Each iteration involves a backward sweep and a forward sweep. In the backward sweep, we solve the Hamilton–Jacobi–Bellman equation for the value function $\phi$ backward in time, from the terminal condition $\phi(T, \cdot)$ to $\phi(0, \cdot)$, using a discrete-time Euler scheme. The interaction term $\int K(\boldsymbol{x}, \boldsymbol{y})\mathrm{d}\rho_t(\boldsymbol{y})$ is approximated by an empirical average over the wholse set of particles for MFG, over a randomly selected subset for Random-MFG, and over a thinned subset produced by KT-SPLIT-Compress for KT-MFG[1] and KT-Compress for KT-MFG[2]. The kernel used for both versions of KT-MFG is a Gaussian kernel with fixed length scale of 1.0. The failure probability is set to be 0.5 for both KT-SPLIT-Compress and KT-Compress. Then, in the forward sweep, we evolve the particles forward in time according to the velocity field $\nabla\phi$ induced by the gradient of the value function. The initial particles are all sampled from $\mathcal{N}(0, 0.1)$. The forward and backward sweeps are alternated for 100 iterations where convergence is empirically observed.

The random batch method [44] is not applicable in this setting, since the Hamilton–Jacobi–Bellman equation is solved

backward in time to obtain $\phi$, followed by a forward-in-time solution of the continuity equation; as a result, it is no longer an interacting particle system.

