# OpenReview forum: "Thinned Mean Field Langevin Dynamics"
_ICML.cc/2026/Conference — ICML 2026 regular_

### Official Review · Reviewer_Rpg1 · 2026-02-21

**Soundness:** 3
**Presentation:** 3
**Significance:** 2
**Originality:** 2
**Overall Recommendation:** 4
**Confidence:** 4

**Summary:**

Mean-field Langevin dynamics (MFLD) solves an entropy-regularized functional over the space of probability measures; however, its discretization is an interacting particle system, which scales quadratically with the number of particles in simulation. The authors propose a new method to speed up MFLD by incorporating techniques from kernel thinning. Roughly speaking, in each round of MFLD, a "well-chosen" coreset of size $O(N^{1/2})$ is used to approximate the interaction component in the gradient. The authors provide convergence of their algorithm and shows it matches the guarantees of MFLD up to logarithmic factors. The authors also provide experiments on canonical applications of MFLD to show the effectiveness of their improvement.

**Compliance With Llm Reviewing Policy:**

Affirmed.

**Final Justification:**

I recommend acceptance because the paper addresses an important issue faced in certain applications of MFLD; the authors also addressed my concerns. The authors should improve presentation on the MFNN experiment and mention limitations of their methods (e.g. their additional term has dependence on the LSI constant, and some applications of MFLD already just cost O(N) runtime per iteration).

**Key Questions For Authors:**

1. As mentioned in the weaknesses section, the MFNN example does not benefit from the thinning procedure. Could the authors please elaborate more on this?
2. I am also concerned with how the runtime cost is presented. It would be more principled to consider varying particles, fixed total number of iterations, and plotting wall-clock time (averaged over say 20 iterations).
3. An alternate method (a conceptually simpler method) for achieving the same speedup is as follows: run $m = \sqrt{N}$ instances with $\sqrt{N}$ particles, then ensemble the particles together. A final optional step is to run the ensembled model for a small number of iterations, say $O(1)$. For instance, see Theorem 3 in [1] for a more principled guarantee on balancing these two factors. Crucially, this approximation bound does *not* incur a dependence on the LSI constant $\bar{\alpha}$. Can the authors discuss this method and provide experiments in this setting?

I will be willing to raise my score to a satisfactory response to the last point because I am concerned on the limited applicability of this thinned method over the method I discussed.

Additional comments:
- There are some missing related works: e.g. trajectory inference [2] and propagation of chaos [3].
- More generally, the authors should provide more discussion on propagation of chaos.

[1] Nitanda et al. Propagation of chaos for mean-field Langevin dynamics and its applications to model ensemble.

[2] Gu et al. Partially observed trajectory inference using optimal transport and a dynamics prior.

[3] Chen et al. Uniform-in-time propagation of chaos for mean field Langevin dynamics.

**Limitations:**

As mentioned above, the authors should clarify that their method only provides improvements for certain MFLD scenarios, which do not include MFNNs. They also should mention the dependence of the LSI constant in their extra error term in Theorem 3.3.

**Strengths And Weaknesses:**

Strengths:
- The paper uses kernel thinning to provide computational speedups for MFLD. Although both components seem standard in the literature, their combination is novel and potentially useful for applications.
- The paper is well-written, and all the theoretical results are well-justified (using standard techniques from prior works).
- The assumptions are verified for the experiments provided.

Weaknesses:
- Training mean-field neural networks (MFNN) has iteration complexity $O(N)$ because there is no interaction term between particles in the objective; this is noted in line 142 also. Thus, I believe there is a mistake in how the results for MFNN are presented in the experiment section.
- More generally, this method will only provide computational advantages when there are interaction terms. Currently, the paper is written as if all MFLD applications suffer this superlinear complexity per iteration; however, this is not true. The paper should be more clear about this.
- While the convergence results are claimed to match the guarantees of non-thinned MFLD, the extra term in Theorem 3.3 $O(\frac{\log^2N}{N\bar{\alpha}})$ scales with the LSI constant $\bar{\alpha}$. This should also be mentioned as a limitation since the LSI is usually exponentially small. This term is only negligible if N is very large (which is not the case in real-world implementations).

---

> ### Author Rebuttal · Authors · 2026-03-28
>
> ## General comments
> We sincerely thank the reviewer for this careful, insightful, and constructive review. We are grateful that the reviewer found the proposed combination “novel and potentially useful for applications,” and that “the theoretical results are well-justified.”
> ## Q1 & W1: Training mean-field neural networks (MFNN) has iteration complexity $O(N)$ because there is no interaction term.
> We thank the reviewer for raising this important point. First of all, strictly speaking, MFNN training **does** involve interaction among particles. To see why, consider a two-layer mean-field network of the form $f_N(z)=\frac{1}{N}\sum_{j=1}^N \Psi(z; x_j)$, where $z$ is the input and $x_1,\dots,x_N$ are parameters. Under the standard squared loss over a dataset $\\{(z\_s,y\_s)\\}\_{s=1}^S$, the gradient with respect to particle $x_i$ is
> $$\frac{2}{S}\sum_{s=1}^S \bigl(f_N(z_s)-y_s\bigr)\frac{1}{N}\nabla_{x_i}\Psi(z_s;x_i) = \frac{2}{SN}\sum_{s=1}^S \left(\frac{1}{N}\sum_{j=1}^N \Psi(z_s;x_j)-y_s\right) \nabla_{x_i}\Psi(z_s;x_i)$$
> Therefore, for the gradient of each $i$-th particle, there indeed exists interactions among particles.
>
> That said, however, in standard neural network training, the per-iteration cost is usually regarded as $O(N)$ instead of $O(N^2)$, because the interactions only appear through the network output $f_N(z\_s)$, which can be computed once and then reused for all particles. This reduces the cost to $O(N)$.
>
> In our experiments, we made a small modification to the MFNN update scheme. Specifically, in the right column of lines 302--305, we use **a different mini-batch of samples to update each $i$-th particle**. This trick makes sure that the network output is different for each $i$-th particle, so the pre-computation trick is no longer applicable, and the $O(N^2)$ cost becomes explicit.
>
> We would like to remark, however, that this is not the standard practical way to train neural networks. Accordingly, this MFNN experiment should be viewed only **as an illustrative setting** to demonstrate the effectiveness of our thinning technique.
> Since MFLD is primarily used as a theoretical model rather than a practical algorithm, we hope this clarification addresses the reviewer’s concern.
>
> We will revise the manuscript to explicitly highlight two points: 1) the quadratic cost for MFNN appears explicitly under our modified update scheme; 2) the MFNN experiment is included as an illustrative example.
>
> ## W3: LSI is usually exponentially small.
> Thank you for highlighting the LSI constant. We agree with the reviewer that it is indeed exponentially small with respect to the dimension [Lemma 5, 72] [ https://arxiv.org/pdf/2409.10440 ]. We will explicitly acknowledge this issue and clarify how it affects the interpretation of our bound.
>
> ## Q2: I am also concerned with how the runtime cost is presented.
> We thank the reviewer for bringing up an alternative way of comparison for our experiments. We agree that, ideally, all methods would be benchmarked under the same wall-clock time. In fact, we follow this protocol in the last two experiments (Figures 4 and 5). For Figures 2 and 3, we instead report computational cost defined in the second column of line 311, since those experiments are relatively small-scale and the runtime differences are less pronounced. We will revise the presentation to make this rationale clearer.
>
> ## Q3: An alternate method
> We thank the reviewer for suggesting this additional baseline, which we refer to below as the Ensemble method. In fact, this baseline is **closely related to the random batch method (RBM) [39]**, which is already described in lines 233-238 and compared extensively in the paper. Specifically, in RBM, the $N$ particles are partitioned at each iteration into $\sqrt{N}$ batches, each of size $\sqrt{N}$, and each batch evolves according to the MFLD update using only within-batch interactions. Thus, there is no interaction across different batches.
>
> The Ensemble baseline is closely related to RBM, yet with two main differences: (1) in RBM, the partition is resampled at each iteration, whereas in the Ensemble method it is fixed throughout training; and (2) the Ensemble method does not include the final training step of the aggregated model. We run this Ensemble baseline in the MFNN setting with final aggregated traiing for 10 steps and provide the results: https://drive.google.com/file/d/1pAiq7JQNGs6Rw12eor76vjBL3A9Jr6Tf/view?usp=sharing.
>
> The results show that the Ensemble method has similar performance as RBM, and is clearly outperformed by our KT-MFLD method before the final aggregated training step. On the other hand, the final training step substantially improves its performance, but also incurs a much higher computational cost due to the $O(N^2)$ interaction cost. We will include this additional figure in the paper and add a more detailed discussion of the relationship between RBM and the reviewer’s suggested Ensemble baseline.

---

> > ### Author Rebuttal · Reviewer_Rpg1 · 2026-03-31
> >
> > Thanks for clarifying my questions regarding the MFNN and experiment details, along with running the additional experiments on the ensemble-based method. It is nice to see that the small LSI doesn't seem to be an issue for the new experiments. Nevertheless, the authors should mention the dependence on the LSI constant in the revision of their paper. I have adjusted my score in light of the response.

---

### Official Review · Reviewer_ihv7 · 2026-03-03

**Soundness:** 3
**Presentation:** 3
**Significance:** 3
**Originality:** 3
**Overall Recommendation:** 5
**Confidence:** 4

**Summary:**

In this paper the authors consider mean field Langevin dynamics (MFLD) for minimizing entropy-regularized functionals of the form $F(\mu) - \sigma \text{ent}(\mu)$ on the space of probability measures over $\mathbb{R}^d$. The gradient flow for this minimization corresponds to a SDE on the population level. When $F$ is not linear, the corresponding drift of this SDE (which is $\nabla F'\[\mu_t\] (X_t)$) depends on the current distribution of the simulated population.
For simulating the gradient flow one typically simulates an ensemble of particles which is then used also to estimate the population distribution. As a consequence, in the naive implementation this leads to a complexity of $~N^2$ for a population size $N$.

In this article, the authors propose to use a specific thinning method (KT-Split-Compress) for MFLD, that is, using only a subset of the ensemble of particles for estimating $\nabla F'(\mu_t) (X_t)$ in order to reduce computational complexity. They prove that the thinning leads to reduced computational effort given a certain fixed required sample accuracy.

**Compliance With Llm Reviewing Policy:**

Affirmed.

**Final Justification:**

I think this is an overall good paper for the following reasons:

* It addresses an interesting, Langevin dynamics where the drift is a function of the current distribution so that an ensamble of particles has to be simulated.
* It solves the main issue, the computational complexity of estimating the drift for each particle, in a good way by using the KT-Split-Compress algorithm.
* The proofs are correct and technically strong.

**Key Questions For Authors:**

1. Around line 150 I do not think the example is properly put into the considered framework of the paper. In particular, in your framework $R_1$ should be a function whose input is solely an expectation over the distribution $\mu$. This is not the case for the example. Please make this more rigorous.
2. In the "illustrative applications" 2 and 3 in both cases you "disregard" a part of the opbjective? Why and how is this interpreted?
4. In line 267 on the right, can you elaborate on the RKHS. In particular, the kernel depends on the sample $z_s$ and is therefore random? Also why is the norm $\||\psi(z_s,\cdot)\||_{H_k}=1$
5. I think there is an error in lemma B.9. Usually, swapping integral and differentiation is argued by Lebesgue's dominated convergence, which relies on an integrable pointwise upper bound of the derivative. Also the cited reference requires uniform integrability of the difference quotients. This is not argued within the article. Moreover, also condition A.2a is not correctly checked. Please clarify/adjust.

6. Smaller remarks:
* Briefly comment on $\rho$ after eq (3)
* In theorem 3.3 the distribution $\pi^{\otimes N}$ is not introduced/explained properly.
* In eq (4) within $R_1'$ use a different letter than $x$ for the integration variable to distinguish from the outside $x$.
* Line 627: "and and" should be "and"
* Line 723: I do not understand this. Propo 2.1 in the main text still leaves $g$ up to choice.
* In proposition B.1 what is the choice of $M$? I assume $\sqrt{n}$
* Please state how you derive line 800
* line 900 the reference to lemma B.4 hould be B.3 I think
* lines 913-914: The notation is poorly chosen as the subscript $t$ for the $x$ variables should be fixed whereas the subscript $t$ for $y$ is the varying time variable.
* line 941 "Gronwell" should be "Grönwall" or "Gronwall"

**Limitations:**

yes

**Strengths And Weaknesses:**

Soundness: The paper seems mostly technically sound. See below for some detailed comments.

Presentation: While the main content is well-presented I found the proofs of the paper to be not specifically well-presented for two reasons. First, the results seemed to be a little chaotically ordered in the appendix, albeit this might be a matter of taste (for instance the ordering Prop B.1 -> Prop B.2 -> Proof of B.1 -> Proof of B.2).
Moreover, the paper relies a lot on results from other articles. However, these results are not recalled but only referenced. While in general, this is perfectly fine, in my opinion due to the amount of cited results it is a little difficult to follow the theoretical arguments. The paper would likely be improved if for instance at the beginning of the proofs section some important results from other works would be recalled verbatim. However, this might be a matter of personal preference.

Significance: The computational complexity of $N^2$ for MFLD seems to be a major obstacle for its more widespread use as far as I have understood so that the results are significant as they provide a (partial) remedy in this regard. In particular, in this regard I want to point out that I found it a positive aspect that the authors provide several concrete application examples of the considered MFLD model.

Originality: The proofs are taken to a large extent from other articles. However, the application of the considered KT-Compress algorithm for thinning in MFLD is creative and original.

---

> ### Author Rebuttal · Authors · 2026-03-28
>
> ## General comments
> We sincerely thank the reviewer for this careful, insightful, and constructive review. We are encouraged by the reviewer’s positive assessment of the paper’s technical soundness, and especially by the recognition that our results provide a “significant and creative partial remedy” to the quadratic complexity of MFLD. We are also very grateful for the reviewer’s detailed and mathematically serious comments, which we are going to address in detail below.
>
> ## Presentation
> Thank you for this helpful suggestion. We will reorganize the appendix so that the statements and proofs appear in a more natural order. We will also restate Theorem 2 in [62] as a lemma, which is the main argument used in the proof, so that the reader can follow the proof without cross-referencing multiple papers.
>
> ## Q1: Around line 150, $R_1$ should be a function whose input is solely an expectation over the distribution $\mu$. This is not the case for the example.
> The first term in the objective $F\_0(\\mu)$ in Eq. (3) is $F\_0(\\mu)=\\mathbb{E}\_{u\\sim\\rho}[R\_1(\\int q\_1(u, x) d \mu(x))]$, where $u$ is considered a latent variable. In our example around line 150, the objective is $\\mathbb{E}\_{(z,y)\\sim\\rho}[(\\int\\Psi(z, x) d \\mu(x) - y)^2]$ (using squared loss for simplicity here). In the example, the latent variable $u$ should be identified with the pair $(z, y)$, so the example fits the definition of $F\_0$ by taking $q\_1(u, x)=\Psi(z, x)$ and $R\_1(t ; u)=(t-y)^2$. We agree, however, that the notation in the current draft is imprecise, since $R_1$ in the example also depends on the latent $u$. We will revise the notation to ensure the example is consistent with the general formulation.
>
> ## Q2: In the "illustrative applications" 2 and 3 in both cases you disregard a part of the objective?
> Our definition of the objective $F_0(\mu)$ is intentionally broad enough to cover all three applications within a single framework. In Application 1, only the first term appears, whereas in Applications 2 and 3, only the second term appears. The quadratic cost of MFLD arises for every term in $F_0(\mu)$ and our thinning technique also applies to either term in $F_0(\mu)$. We use the unified formulation of $F_0(\mu)$ to avoid repeating the same analysis across separate applications.
>
> ## Q3:  In line 267 on the right, can you elaborate on the RKHS.
>
> For each set of fixed sample $\\{z\_s\\}\_{s=1}^n$, the kernel is $k(x,x^\circ)=\\sum_{s=1}^n \Psi(z_s,x)\Psi(z_s,x^\circ)$. So you are right that the kernel depends on the sample. However, throughout the paper, we work conditionally on the observed data samples, so once $\\{z_s\\}\_{s=1}^n$ is fixed, the kernel is deterministic.
>
> The functions in the associated RKHS of this kernel are of the form $f(\\cdot)=\\sum\_{s=1}^n a\_s \Psi(z\_s,\\cdot)$,
> equipped with the inner product $\\langle\\sum\_{s=1}^n a\_s \\Psi(z\_s,\\cdot), \\sum\_{s=1}^n b\_s \\Psi(z\_s,\\cdot)\\rangle\_{\\mathcal{H}\_k}=\\sum\_{s=1}^n a\_s b\_s.$
> Therefore $\\|\\Psi(z\_s,\\cdot)\\|\_{\\mathcal{H}\_k}^2=\\langle\\Psi(z\_s,\\cdot),\\Psi(z\_s,\\cdot)\\rangle\_{\\mathcal{H}\_k}=1$.
> We will make this interpretation more explicit in the camera-ready version of the paper.
>
> ## Q4: I think there is an error in lemma B.9.
> It is a good catch, and we appreciate the reviewer’s careful reading of the proof in the Appendix. In Lemma B.9, our original assumptions only guarantee pointwise integrability of the derivative, i.e, $\int|\varkappa(y, y^{\prime})| \|\nabla_{x} p_{x}(y)\| d y<\infty$ for any $x$, but what is needed to swap integral and differentiation is uniform integrability over a small neighbourhood around every $x\in\mathbb{R}^d$. In the revised manuscript, we will address this point by imposing a stronger sufficient condition in Lemma B.9, namely $\int|\varkappa(y, y^{\prime})| \sup\_x\|\nabla_{x} p_{x}(y)\|d y<\infty$.
>
> Regarding the reviewer’s comment on condition A.2a, we again only established pointwise integrability, whereas A.2a requires a uniform integrability. Fortunately, this condition is straightforward to verify in our setting because $\varkappa$ is bounded and $p_x(y)$ is a probability density. Indeed, for any $y'$, $\\sup\_x\\int |\\varkappa(y,y') p\_x(y)| dy\\leq \\|\\varkappa\\|\_{\\infty}\\sup\_x\\int p\_x(y) dy=\|\varkappa\|_{\infty}<\infty$. We will revise the proof in the camera-ready version of the paper.
>
> ## Q5: Minor comments
> Thank you for the very careful reading and for catching these issues. We will correct them in the camera-ready version of the paper. For completeness, we briefly clarify some of the points. In Proposition B.1, yes, the correct choice is $M=\sqrt{n}$. Line 800 is obtained by rewriting the high-probability bound in terms of $u=\log(1/\delta)$, and then inverting the resulting relation between $t$ and $u$ to derive the corresponding tail bound.

---

> > ### Author Rebuttal · Reviewer_ihv7 · 2026-04-01
> >
> > The authors responded to all my questions. (Although the response to Q1 was actually just a repetition of what is anyway in the manuscript. However, Q1 should not be too big of an issue)

---

### Official Review · Reviewer_yGE6 · 2026-03-10

**Soundness:** 3
**Presentation:** 2
**Significance:** 3
**Originality:** 3
**Overall Recommendation:** 4
**Confidence:** 4

**Summary:**

This paper accelerates Mean Field Langevin Dynamics by using kernel thinning to select a smaller representative subset of particles at each iteration, reducing the per-step cost from O(N²) to O(N^(3/2)) while maintaining an O(N⁻¹(log N)²) approximation error.

**Compliance With Llm Reviewing Policy:**

Affirmed.

**Final Justification:**

The author's reponses to my questions have been addressed well. However, I can not assese the revised abstract, introduction (Weakness 2) for ICML's policy.

**Key Questions For Authors:**

(1) Can you provide evidence that Assumption 3.2 holds for the neural network application?

(2) Since the representative particles are just a subset of existing ones and cannot move freely, how do you characterize the risk of error accumulation over many iterations?

Please c.f. "Weakness" for details.

**Limitations:**

Yes, the limitations are adequately discussed.

**Strengths And Weaknesses:**

### Strengths

The theoretical analysis showing that thinning achieves nearly the same error rate as full MFLD is solid and clearly improves over random subsampling.


### Weaknesses

(1) The main theorem relies on Assumption 3.2, which requires certain functions to belong to a specific RKHS. It is unclear whether this actually holds for neural networks (to our knowledge, not, so please clarify this).

(2) The term "thinned" is never explained in the abstract or introduction and the introduction is complicated, making the paper harder to follow than necessary (although i'm familiar with Langevin dynamic).

(3) While KT-Split-Compress selects subsets well, it cannot compensate for poorly distributed particles. This creates a risk of error accumulation over time that the per-step analysis does not address.

(4) The core idea resembles inducing points in sparse Gaussian processes, but here the representative particles are only selected from the current set rather than optimized globally, which seems limiting.

---

> ### Author Rebuttal · Authors · 2026-03-28
>
> ## General comments
> We sincerely thank the reviewer for the careful reading of our manuscript. We are pleased that the main message of the paper was clearly conveyed, namely that our method can “reduce the per-step cost from $\mathcal{O}(N^2)$ to $\mathcal{O}(N^{3/2})$ while maintaining an $\mathcal{O}(N^{-1}(\log N)^2)$ approximation error”. We are glad to see that you find “the theoretical analysis solid and clearly improves over random subsampling.”
> ## W1 & Q1: The main theorem relies on Assumption 3.2, which requires certain functions to belong to a specific RKHS. It is unclear whether this actually holds for neural networks (to our knowledge, not, so please clarify this).
> Assumption 3.2 actually holds for neural networks. As we have shown in the first bullet point of Remark 3.4, the neural network function $\Psi$ belongs to a RKHS associated with a finite-rank kernel $k\left(\boldsymbol{x}, \boldsymbol{x}^{\circ}\right)=\sum_{s=1}^n \Psi\left(z_s, \boldsymbol{x}\right) \Psi\left(z_s, \boldsymbol{x}^{\circ}\right)$. We refer to our response to reviewer ihv7, Q3, for a more detailed explanation.
> That said, this is not the kernel we use for KT in the neural network example in the experiments, since it involves an empirical average over the samples and is therefore computationally expensive. Instead, in our first experiment we use a Sobolev kernel; see line 314-318 and the definition of the kernel in Equation (C.1). The RKHS associated with this kernel is the Sobolev space, and the neural network function also typically belongs to that space with a specific smoothness order dependent on the activation function; see, for example, https://arxiv.org/pdf/2508.05141v1. We agree that this results in a distinction between the idealized kernel in the theory and the kernel used in practice, and we will clarify this point further in the revised paper.
> ## Q2 & W3 & W4: Since the representative particles are just a subset of existing ones and cannot move freely, how do you characterize the risk of error accumulation over many iterations? This creates a risk of error accumulation over time that the per-step analysis does not address.
> Thank you for the question. We prove that in Proposition B.2 that the error of the particle system is contractive per iteration up to the approximation error of using a subset of representative particles. Based on Proposition B.2, we next prove in Theorem 3.3 that the overall error **after many iterations** can still be controlled (See the detailed discussion below Theorem 3.3). Regarding the reviewer’s concern that our representative particles cannot move freely, we would like to clarify that our goal is not trying to find the best possible $\mathcal{O}(\sqrt{N})$ particle approximation of some target distribution $\mu$, as is required in the sparse GP setting. This is a substantially more challenging task. Rather, our objective is to identify a subset of size $\mathcal{O}(\sqrt{N})$ that accurately approximates the vector field of the original full $N$-particle system. For this purpose, a subset of the existing $N$ particles constructed via kernel thinning is sufficient, as established in Theorem 3.3.
>
> ## W4: The term "thinned" is never explained in the abstract or introduction and the introduction is complicated, making the paper harder to follow than necessary.
> Yes, we agree. We will revise the abstract and introduction to explain the concrete meaning of “thinning” more clearly at an early stage.

---

> > ### Author Rebuttal · Reviewer_yGE6 · 2026-04-02
> >
> > Thank you for resolving my concerns. I tend to raise my score from 3 to 4.

---

### Decision · Program_Chairs · 2026-04-30

**Decision:**

Accept (regular)

**Comment:**

The paper addresses an important bottleneck in certain applications of mean-field Langevin dynamics by introducing a principled thinning scheme that reduces the per-iteration cost while preserving essentially the same convergence guarantees. The core idea is novel and technically well executed, the theoretical analysis is good, and the paper supports the theory with several concrete applications demonstrating practical utility in settings where interaction costs are the main obstacle. The rebuttal also satisfactorily addressed the main reviewer concerns.

That said, the final version should more clearly acknowledge the key limitations. In particular, the additional error term depends on LSI-constant, which can be very small, and the computational advantage is not universal, as mean-field Langevin dynamics already admit effectively $O(N)$ per-iteration cost in standard mean-field neural network settings. With these caveats made explicit, I believe this is a solid and worthwhile contribution.